# GCondNet: A Novel Method for Improving Neural Networks on Small High-Dimensional Tabular Data

**Andrei Margeloiu**                                                      *am2770@cam.ac.uk*
*Department of Computer Science and Technology*
*University of Cambridge, UK*

**Nikola Simidjievski**                                                   *ns779@cam.ac.uk*
*Precision Breast Cancer Institute, Department of Oncology, University of Cambridge, UK*
*Department of Computer Science and Technology, University of Cambridge, UK*

**Pietro Liò**                                                            *pl219@cam.ac.uk*
*Department of Computer Science and Technology*
*University of Cambridge, UK*

**Mateja Jamnik**                                                        *mj201@cam.ac.uk*
*Department of Computer Science and Technology*
*University of Cambridge, UK*

**Reviewed on OpenReview:** *https://openreview.net/forum?id=yObOH1ndGQ*

## Abstract

Neural networks often struggle with high-dimensional but small sample-size tabular datasets. One reason is that current weight initialisation methods assume independence between weights, which can be problematic when there are insufficient samples to estimate the model's parameters accurately. In such small data scenarios, leveraging additional structures can improve the model's performance and training stability. To address this, we propose GCondNet, a general approach to enhance neural networks by leveraging implicit structures present in tabular data. We create a graph between samples for each data dimension, and utilise Graph Neural Networks (GNNs) to extract this implicit structure, and for conditioning the parameters of the first layer of an underlying predictor network. By creating many small graphs, GCondNet exploits the data's high-dimensionality, and thus improves the performance of an underlying predictor network. We demonstrate GCondNet's effectiveness on 12 real-world datasets, where it outperforms 14 standard and state-of-the-art methods. The results show that GCondNet is a versatile framework for injecting graph-regularisation into various types of neural networks, including MLPs and tabular Transformers. Code is available at `https://github.com/andreimargeloiu/GCondNet`.

## 1 Introduction

Tabular datasets are ubiquitous in scientific fields such as medicine (Meira et al., 2001; Balendra & Isaacs, 2018; Kelly & Semsarian, 2009), physics (Baldi et al., 2014; Kasieczka et al., 2021), and chemistry (Zhai et al., 2021; Keith et al., 2021). These datasets often have a limited number of samples but a large number of features for each sample. This is because collecting many samples is often costly or infeasible, but collecting many features for each sample is relatively easy. For example, in medicine (Schaefer et al., 2020; Yang et al., 2012; Gao et al., 2015; Iorio et al., 2016; Garnett et al., 2012; Bajwa et al., 2016; Curtis et al., 2012; Tomczak et al., 2015), clinical trials targeting rare diseases often enrol only a few hundred patients at most. Despite the small number of participants, it is common to gather extensive data on each individual, such as measuring thousands of gene expression patterns. This practice results in small-size datasets that are high-dimensional,

with the number of features ($D$) greatly exceeding the number of samples ($N$). Making effective inferences from such datasets is vital for advancing research in scientific fields.

When faced with high-dimensional tabular data, neural network models struggle to achieve strong performance (Liu et al., 2017; Feng & Simon, 2017), partly because they encounter increased degrees of freedom, which results in overfitting, particularly in scenarios involving small datasets. Despite transfer learning's success in image and language tasks (Tan et al., 2018), a general transfer learning protocol is lacking for tabular data (Borisov et al., 2022), and current methods assume shared features (Levin et al., 2023) or large upstream datasets (Wang & Sun, 2022; Nam et al., 2022), which is unsuitable for our scenarios. Consequently, we focus on improving training neural networks from scratch.

Previous approaches for training models on small sample-size and high-dimensional data constrained the model's parameters to ensure that similar features have similar coefficients, as initially proposed in (Li & Li, 2008) for linear regression and later extended to neural networks (Ruiz et al., 2023). For applications in biomedical domains, such constraints can lead to more interpretable identification of genes (features) that are biologically relevant (Li & Li, 2008). However, these methods require access to external application-specific knowledge graphs (e.g., gene regulatory networks) to obtain feature similarities, which provide "explicit relationships" between features. But numerous tasks do not have access to such application-specific graphs. We aim to integrate a similar inductive bias, posing that performance is enhanced when similar features have similar coefficients. We accomplish this *without* relying on "explicit relationships" defined in external application-specific graphs.

We propose a novel method GCondNet (**G**raph-**Cond**itioned **Net**works) to enhance the performance of various neural network predictors, such as Multi-layer Perceptrons (MLPs). The key innovation of GCondNet lies in leveraging the "implicit relationships" between *samples* by performing "soft parameter-sharing" to constrain the model's parameters in a principled manner, thereby reducing overfitting. Prior work has shown that such relationships between samples can be beneficial (Fatemi et al., 2021; Kazi et al., 2022; Zhou et al., 2022). These methods, however, typically generate and operate with one graph between samples while relying on additional dataset-specific assumptions such as the smoothness assumption (for extended discussion see Section 4). In contrast, we leverage *sample-wise multiplex graphs*, a novel and general approach to identify and use these potential relationships between samples by constructing many graphs between samples, one for each feature. We then use Graph Neural Networks (GNNs) to extract any implicit structure and condition the parameters of the first layer of an underlying predictor MLP network. Note that GCondNet still considers the samples as independent and identically distributed (IID) at both train-time and test-time because the information from the graphs is encapsulated within the model parameters and is not used directly for prediction (see Section 2.2).

We introduce two similarity-based approaches for constructing the sample-wise multiplex graphs from any tabular dataset. Both approaches generate a graph for each feature in the dataset (resulting in $D$ graphs), with each node representing a sample (totalling $N$ nodes per graph). For instance, in a gene expression dataset, we create a unique graph of patients for each gene. Unlike other methods (Ruiz et al., 2023; Li & Li, 2008; Scherer et al., 2022) that require external knowledge for constructing the graphs, our graphs can be constructed from any tabular dataset. We also propose a decaying mechanism which improves the model's robustness when incorrect relationships between samples are specified.

The inductive bias of GCondNet lies in constraining the model's parameters to ensure similar features have similar coefficients at the beginning of training, and we show that our approach yields improved downstream performance and enhanced model robustness. One reason is that creating many small graphs effectively "transposes" the problem and makes neural network optimisation more effective because we leverage the high-dimensionality of the data to our advantage by generating many small graphs. These graphs serve as a large training set for the GNN, which in turn computes the parameters of the MLP predictor. In addition, our approach also models a different aspect of the problem – the structure extracted from the implicit relationships between samples – which we show serves as a regularisation mechanism for reducing overfitting.

Our contributions are summarised as follows:

1. We propose a novel method, GCondNet, for leveraging implicit relationships between samples into neural networks to improve predictive performance on small sample-size and high-dimensional tabular data. Our

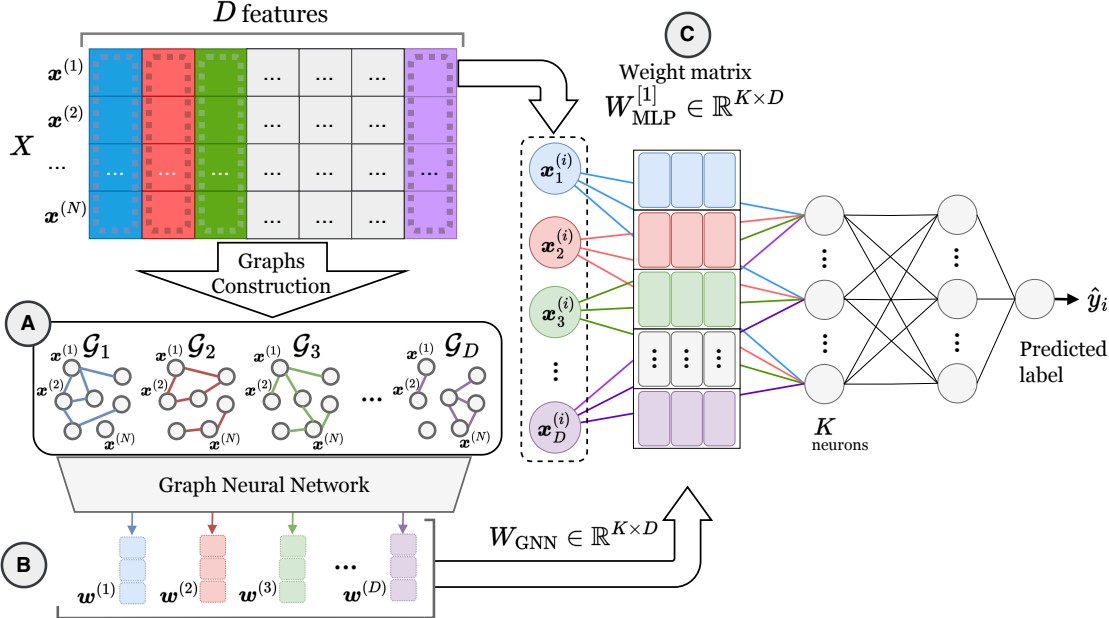

Figure 1: GCondNet is a general method for leveraging *implicit* relationships between samples to improve the performance of *any* predictor network with a linear first layer, such as a standard MLP, on tabular data. **(A)** Given a tabular dataset $\boldsymbol{X} \in \mathbb{R}^{N \times D}$, we generate a graph $\mathcal{G}_j$ for each feature in the dataset (results in $D$ graphs), with each node representing a sample (totalling $N$ nodes per graph). **(B)** The resulting graphs are passed through a shared Graph Neural Network (GNN), which extracts graph embeddings $\boldsymbol{w}^{(j)} \in \mathbb{R}^K$ from each graph $\mathcal{G}_j$. We concatenate the graph embeddings into a matrix $\boldsymbol{W}_{\mathrm{GNN}} = [\boldsymbol{w}^{(1)}, ..., \boldsymbol{w}^{(D)}]$. **(C)** We use $\boldsymbol{W}_{\mathrm{GNN}}$ to parameterise the first layer $\boldsymbol{W}_{\mathrm{MLP}}^{[1]}$ of the MLP predictor as a convex combination $\boldsymbol{W}_{\mathrm{MLP}}^{[1]} = \alpha \boldsymbol{W}_{\mathrm{GNN}} + (1 - \alpha) \boldsymbol{W}_{\mathrm{scratch}}$, where $\boldsymbol{W}_{\mathrm{scratch}}$ is initialised to zero.

    method is general and can be applied to any such tabular dataset, unlike other methods that rely on external application-specific knowledge graphs.

2. We validate GCondNet's effectiveness across 12 real-world biomedical datasets. We show that for such datasets, our method consistently outperforms an MLP with the same architecture and, in fact, outperforms all 14 state-of-the-art methods we evaluate.

3. We analyse GCondNet's inductive bias, showing that our proposed *sample-wise multiplex graphs* improve performance and serve as an additional regularisation mechanism. Lastly, we demonstrate that GCondNet is robust to various graph construction methods, which might also include incorrect relationships.

## 2 Method

**Problem formulation.** We study tabular classification problems (although the method can be directly applied to regression too), where the data matrix $\boldsymbol{X} := [\boldsymbol{x}^{(1)}, ..., \boldsymbol{x}^{(N)}]^\top \in \mathbb{R}^{N \times D}$ comprises $N$ samples $\boldsymbol{x}^{(i)} \in \mathbb{R}^D$ of dimension $D$, and the labels are $\boldsymbol{y} := [y_1, ..., y_N]^\top$.

**Method overview.** Our method applies to *any* network with a linear layer connected to the input features, and for illustration, we assume an MLP predictor network. Figure 1 presents our proposed method, which has two components: (i) a predictor network (e.g., MLP) that takes as input a sample $\boldsymbol{x}^{(i)} \in \mathbb{R}^D$ and outputs the predicted label $y_i$; and (ii) a Graph Neural Network (GNN) that takes as input *fixed* graphs ($D$ of them) and generates the parameters $\boldsymbol{W}_{\mathrm{MLP}}^{[1]}$ for the first input layer of the predictor MLP network. Note that the GNN is shared across graphs. Since these graphs are fixed for all inputs $\boldsymbol{x}^{(i)}$, GCondNet maintains the dimensionality of the input space (which remains $D$).

---

**Algorithm 1** Computing $\boldsymbol{W}_{\text{GNN}}$.

---

1: **for** each feature $j = 1, 2, \ldots, D$ **do**
2:     node-embeddings $= \text{GNN}(\mathcal{V}_j, \mathcal{E}_j)$     ▷ $\mathcal{V}_j$ represents the nodes, and $\mathcal{E}_j$ represents the edges of the $j$-th graph
3:     $\boldsymbol{w}^{(j)} = f_{\text{agg}}(\text{node-embeddings})$  ▷ Aggregate all node embeddings to obtain the graph embedding $\boldsymbol{w}^{(j)} \in \mathbb{R}^K$
4: **end for**
5: $\boldsymbol{W}_{\text{GNN}} = [\boldsymbol{w}^{(1)}, \ldots, \boldsymbol{w}^{(D)}]$                    ▷ Concatenate the graph embeddings

---

In particular, we parameterise the MLP's first layer $\boldsymbol{W}_{\text{MLP}}^{[1]}$ as a convex combination of a weight matrix $\boldsymbol{W}_{\text{GNN}}$ generated by the GNN (by extracting the implicit structure between samples), and a weight matrix $\boldsymbol{W}_{\text{scratch}}$ initialised to zero:

$$\boldsymbol{W}_{\text{MLP}}^{[1]} := \alpha \boldsymbol{W}_{\text{GNN}} + (1 - \alpha) \boldsymbol{W}_{\text{scratch}} \tag{1}$$

The mixing coefficient $\alpha$ determines how much the model should be conditioned on the relationships between samples learnt by the GNN. We schedule $\alpha$ to linearly decay $1 \to 0$ over $n_\alpha$ training steps, as further motivated in Section 2.2. We found that GCondNet is robust to $n_\alpha$, as supported by the statistical tests in Appendix F.1.

**Computing $\boldsymbol{W}_{\text{GNN}}$.** We train the GNN model concurrently with the MLP predictor to compute the weight matrix $\boldsymbol{W}_{\text{GNN}}$. To do this, we use the training split of $\boldsymbol{X}$ to generate a graph $\mathcal{G}_j = (\mathcal{V}_j, \mathcal{E}_j)$ for each feature in the dataset (resulting in $D$ graphs), with each node representing a sample (totalling $N$ nodes per graph). For example, in a gene expression dataset, one graph of patients is created for each gene. All graphs are simultaneously passed to the GNN and are *fixed* during the training process. This way, we take advantage of high-dimensional data by creating many graphs to train the GNN. We describe the graph construction in Section 2.1 and investigate the impact of this choice in Section 3.2.

At each training iteration, we use the GNN to extract graph embeddings from these graphs, as presented in Algorithm 1. For each of the $D$ graphs, we first apply the GNN to obtain node embeddings of size $K$ for all nodes. We then compute graph embeddings $\boldsymbol{w}^{(j)} \in \mathbb{R}^K$ by using a permutation invariant function $f_{\text{agg}}$ to aggregate the node embeddings. Thus, the graph embeddings are also of size $K$, which is independent of the number of nodes in the graphs. These embeddings are then concatenated horizontally to form the weight matrix $\boldsymbol{W}_{\text{GNN}} = [\boldsymbol{w}^{(1)}, \boldsymbol{w}^{(2)}, \ldots, \boldsymbol{w}^{(D)}]$. Finally, we use the resulting matrix $\boldsymbol{W}_{\text{GNN}}$ to parameterise the first layer of the underlying MLP predictor network that outputs the final prediction, as shown in Equation 1. Appendix A presents the complete pseudocode for training GCondNet.

**Test-time inference.** We emphasise that the GNN and the associated graphs are employed exclusively during the training phase, becoming obsolete once the mixing coefficient $\alpha$ reaches zero after $n_\alpha$ training steps. The predictor MLP retains its final weights upon training completion, rendering the GNN and graphs unnecessary for test inference. Test input samples are exclusively processed by the predictor MLP – resulting in a model size and inference speed identical to a standard MLP.

## 2.1 Sample-wise Multiplex Graphs

We propose a novel and general graph construction method from tabular data, creating a multiplex graph $\mathcal{G} = \{\mathcal{G}_1, \ldots, \mathcal{G}_D\}$, where each graph layer $\mathcal{G}_j = (\mathcal{V}_j, \mathcal{E}_j)$ represents the relations across feature $j$ and is constructed using *only* the values $\boldsymbol{X}_{:,j}$ of that feature. This enables the use of simple distance metrics between nodes and eliminates the need to work in a high-dimensional space where distances can be inaccurate. Note that, the graph construction phase is typically fast, performed once per dataset and adds a negligible computational overhead.

**Node features.** The nodes $\mathcal{V}_j$ of each graph represent the $N$ training samples. The node features are one-hot encoded vectors, with the feature value for a sample located in the corresponding position in the one-hot encoding. For instance, if the values of feature $j$ of three training samples $\boldsymbol{x}^{(1)}, \boldsymbol{x}^{(2)}, \boldsymbol{x}^{(3)}$ are $\boldsymbol{X}_{:,j} = [\boldsymbol{x}_j^{(1)}, \boldsymbol{x}_j^{(2)}, \boldsymbol{x}_j^{(3)}]$, then the first node's features would be $[\boldsymbol{x}_j^{(1)}, 0, 0]$, the second node's features would be $[0, \boldsymbol{x}_j^{(2)}, 0]$, and the third node's features would be $[0, 0, \boldsymbol{x}_j^{(3)}]$.

**Edges.** We propose two similarity-based methods for constructing the edges between samples from tabular data, which assume that similar samples should be connected. To measure sample similarity, we calculate the $\ell_1$ distances between the feature values $\boldsymbol{X}_{:,j}$. Using the earlier example, if the feature values used to create one graph are $\boldsymbol{X}_{:,j} = [\boldsymbol{x}_j^{(1)}, \boldsymbol{x}_j^{(2)}, \boldsymbol{x}_j^{(3)}]$, then the distances between nodes (based on which we create edges) are $\|\boldsymbol{x}_j^{(1)} - \boldsymbol{x}_j^{(2)}\|_1$, $\|\boldsymbol{x}_j^{(1)} - \boldsymbol{x}_j^{(3)}\|_1$, and $\|\boldsymbol{x}_j^{(2)} - \boldsymbol{x}_j^{(3)}\|_1$. While in this paper we use $\ell_1$ distance, other suitable distance functions can also be applied.

The two types of edges are: (i) **KNN graphs** connect each node with the closest $k$ neighbours (we set $k = 5$ in this paper). The memory complexity is $O(D \cdot N \cdot K)$, enabling GCondNet to scale linearly memory-wise w.r.t. both the sample size $N$ and the number of features $D$. The time complexity for KNN graph construction is $O(D \cdot N \log N + D \cdot N \cdot K)$, with the first term for sorting features and the second for creating edges. (ii) **Sparse Relative Distance (SRD) graphs** connect a sample to all samples with a feature value within a specified distance. This process creates a network topology where nodes with common feature values have more connections, and we use an accept-reject step to sparsify the graph (all details are included in Appendix B). The time complexity mirrors that of the KNN graphs. For smaller datasets, the number of edges in the SRD graphs is comparable to those of KNN graphs. In larger datasets, KNN graphs are likely more scalable due to direct control over the number of edges via $K$.

## 2.2 Rationale for Model Architecture

**The inductive bias of GCondNet** ensures that *similar features have similar weights in the first layer of the NN* at the beginning of training. Uniquely to GCondNet, the feature similarity is uncovered by training GNNs end-to-end on graphs defining the implicit relationships between samples across each feature. Thus, our approach ultimately learns the feature similarity by looking at the relationships between samples. For example, if features $i$ and $j$ have similar values across samples, they define similar graphs, leading to similar graph embeddings $\boldsymbol{w}^{(i)}$ and $\boldsymbol{w}^{(j)}$. These embeddings correspond to the first layer weights $\boldsymbol{W}_{\text{MLP}}^{[1]}$ in the neural network.

**GCondNet is appropriate when** $D \gg N$ because it introduces a suitable inductive bias that enhances model optimisation, as we demonstrate in Section 3. On small sample-size and high-dimensional data, conventional neural approaches (such as an MLP) tend to exhibit unstable training behaviour and/or converge poorly – one reason is a large degree of freedom for the small datasets. This happens because; (i) the number of parameters in the first layer is proportional to the number of features; and (ii) modern weight initialisation techniques (Glorot & Bengio, 2010; He et al., 2015) assume independence between the parameters within a layer. Although the independence assumption may work well with large datasets, as it allows for flexibility, it can be problematic when there are too few samples to estimate the model's parameters accurately (as we show in Section 3.2). GCondNet is designed to mitigate these training instabilities: (i) by constraining the model's degrees of freedom via an additional GNN that outputs the model's first layer, which includes most of its learning parameters; and (ii) by providing a more principled weight initialisation on the model's first layer (because at the start we have $\boldsymbol{W}_{\text{MLP}}^{[1]} = \boldsymbol{W}_{\text{GNN}}$).

**We parameterise the first layer** due to its large number of parameters and propensity to overfit. On high-dimensional tabular data, an MLP's first layer holds most parameters; for instance, on a dataset of 20,000 features, the first layer has 98% of the parameters of an MLP with a hidden size 100.

**GCondNet still consider the samples as IID** at both train-time and test-time. Recall that samples are IID if they are independent, conditioned on the model's parameters $\theta$ (Murphy, 2022), so that $p\left(y_1, y_2 \mid \boldsymbol{x}^{(1)}, \boldsymbol{x}^{(2)}, \theta\right) = p\left(y_1 \mid \boldsymbol{x}^{(1)}, \theta\right) \cdot p\left(y_2 \mid \boldsymbol{x}^{(2)}, \theta\right)$. Note that unlike distance-based models (e.g., KNN), our graphs are *not* used directly to make predictions. In GCondNet, to make a prediction, input samples are exclusively processed by the predictor MLP, which uses the same model parameters $\theta$ across all samples. Because all information extracted from our sample-wise graphs is encapsulated within the model parameters $\theta$, the above IID equation holds for GCondNet.

**In terms of graph construction**, the conventional approach would be to have one large graph where nodes are features and $\boldsymbol{w}^{(j)}$ are node embeddings. In contrast, we generate many small graphs and compute $\boldsymbol{w}^{(j)}$ as graph embeddings. Our approach offers several advantages: (i) Having multiple graphs "transposes" the problem and uses the high-dimensionality of the data to our advantage by generating many small graphs

which serve as a large training set for the GNN. (ii) It allows using simple distance metrics because the nodes contain only scalar values; in contrast, taking distances between features would require working in a high-dimensional space and encountering the curse of dimensionality. (iii) The computation is efficient because the graphs are small due to small-size datasets. (iv) Flexibility, as it can incorporate external knowledge graphs, if available, by forming hyper-graphs between similar feature graphs.

**Decaying the mixing coefficient** $\alpha$ introduces flexibility in the learning process by enabling the model to start training initialised with the GNN-extracted structure and later adjust the weights more autonomously as training advances. Since the true relationships between samples are unknown, the GNN-extracted structure may be noisy or suboptimal for parameterising the model. At the start, $\alpha = 1$ and the first layer is entirely determined by the GNN ($\boldsymbol{W}_{\mathrm{MLP}}^{[1]} = \boldsymbol{W}_{\mathrm{GNN}}$). After $\alpha$ becomes 0, the model trains as a standard MLP (the GNN is disabled), but its parameters have been impacted by our proposed method and will reach a distinct minimum (evidenced in Section 3.1 by GCondNet consistently outperforming an equivalent MLP). In contrast to our decaying of the mixing coefficient $\alpha$, maintaining $\alpha$ fixed (similar to PLATO's (Ruiz et al., 2023) inductive bias) leads to unstable training (see our experiments in Section 3.2). Moreover, if $\alpha$ was fixed, it would need optimisation like other hyperparameters, while by decaying $\alpha$ we avoid this time-consuming tuning.

## 3 Experiments

Our central hypothesis is that exploiting the implicit sample-wise relationships by performing soft parameter-sharing improves the performance of neural network predictors. First, we evaluate our model against 14 benchmark models (Section 3.1). We then analyse the inductive bias of our method (Section 3.2), its effect on optimisation, and GCondNet's robustness to different graph construction methods.

**Datasets.** We focus on classification tasks using small-sample and high-dimensional datasets and consider 12 real-world tabular biomedical datasets ranging from $72 - 200$ samples, and $3312 - 22283$ features. We specifically keep the datasets small to mimic practical scenarios where data is limited. See Appendix C for details on the datasets.

**Evaluation.** We evaluate all models using a 5-fold cross-validation repeated 5 times, resulting in 25 runs per model. We report the mean $\pm$ std of the test balanced accuracy averaged across all 25 runs. To summarise the results in the manuscript, we rank the methods by their predictive performance. For each dataset, methods are ranked from 1 (the best) to 12 (the worst) based on their mean accuracy. If two methods have the same accuracy (rounded to two decimals), they obtain the same per-dataset rank. The final rank for each method is the average of their per-dataset ranks, which may be a non-integer.

**GCondNet architecture and settings**. GCondNet uses an MLP predictor model (as its backbone) with three layers with $100, 100, 10$ neurons. The GNN within GCondNet is a two-layer Graph Convolutional Network (GCN) (Kipf & Welling, 2017). The permutation invariant function $f_{\mathrm{agg}}$ for computing graph embeddings is global average pooling[1]. We decay the mixing coefficient $\alpha$ over $n_\alpha = 200$ training steps, although we found that GCondNet is robust to the number of steps $n_\alpha$, as supported by the statistical tests in Appendix F.1. We present the results of GCondNet with both KNN and SRD graphs. We provide complete reproducibility details for all methods in Appendix D, and the training times for GCondNet and other methods are in Appendix E.

**Benchmark methods.** We evaluate 14 benchmark models, encompassing a standard MLP and modern methods typically employed for small sample-size and high-dimensional datasets, such as DietNetworks (Romero et al., 2017), FsNet (Singh & Yamada, 2023), SPINN (Feng & Simon, 2017), DNP (Liu et al., 2017), and WPFS (Margeloiu et al., 2023), all of which use the same architecture as GCondNet for a fair comparison. We also include contemporary neural architectures for tabular data, like TabNet (Arık & Pfister, 2021), Tab-Transformer (Huang et al., 2020), Concrete Autoencoders (CAE) (Balın et al., 2019), and LassoNet[2] (Lemhadri et al., 2021), and standard methods such Random Forest (Breiman, 2001) and LightGBM (Ke et al., 2017).

We also compare the performance of GCondNet with GNNs on tabular data where relationships between samples are not explicitly provided. In particular, we evaluate Graph Convolutional Network (GCN) (Kipf &

---

[1]Using hierarchical pooling (Ying et al., 2018; Ranjan et al., 2020) led to unstable training and significantly poorer performance.
[2]We discuss LassoNet training instabilities in Appendix D.

Table 1: **Overall, GCondNet outperforms other benchmark models.** We show the classification performance of GCondNet with KNN and SRD graphs and 14 benchmark models on 12 real-world datasets. $N/D$ represents the per-dataset ratio of samples to features. We report the mean ± std of the test balanced accuracy averaged over the 25 cross-validation runs. We highlight the **First**, **Second** and **Third** ranking accuracy for each dataset. To aggregate the results, we also compute each method's average rank across datasets, where a higher rank implies higher accuracy in general. Overall, GCondNet ranks best and generally outperforms all other benchmark methods.

| Dataset | gli | smk | allaml | cll | glioma | prostate | toxicity | tcga-survival | tcga-tumor | meta-dr | meta-p50 | lung | Avg. |
| $N/D$ | 0.004 | 0.009 | 0.01 | 0.01 | 0.011 | 0.017 | 0.03 | 0.046 | 0.046 | 0.048 | 0.048 | 0.059 | Rank |
|---|---|---|---|---|---|---|---|---|---|---|---|---|---|
| DietNetworks | $76.42_{\pm13.2}$ | $62.71_{\pm9.4}$ | $92.00_{\pm8.4}$ | $68.84_{\pm9.2}$ | $68.00_{\pm14.8}$ | $81.71_{\pm11.0}$ | $82.13_{\pm7.4}$ | $53.62_{\pm5.5}$ | $46.69_{\pm7.1}$ | $56.98_{\pm8.7}$ | $95.02_{\pm4.8}$ | $90.43_{\pm6.2}$ | 8.92 |
| FsNet | $74.52_{\pm11.7}$ | $56.27_{\pm9.2}$ | $78.00_{\pm12.9}$ | $66.38_{\pm9.2}$ | $53.17_{\pm12.9}$ | $84.74_{\pm9.8}$ | $60.26_{\pm8.1}$ | $53.83_{\pm7.9}$ | $45.94_{\pm9.8}$ | $56.92_{\pm10.1}$ | $83.86_{\pm8.2}$ | $91.75_{\pm3.0}$ | 11.17 |
| DNP | $83.17_{\pm12.1}$ | $66.61_{\pm8.4}$ | $96.18_{\pm5.7}$ | $85.13_{\pm5.5}$ | $75.00_{\pm12.8}$ | $88.71_{\pm6.8}$ | $93.49_{\pm6.2}$ | $58.14_{\pm8.2}$ | $47.53_{\pm8.7}$ | $55.79_{\pm7.1}$ | $93.56_{\pm5.5}$ | $92.81_{\pm6.7}$ | 5.58 |
| SPINN | $83.39_{\pm9.8}$ | $65.91_{\pm7.6}$ | $96.78_{\pm6.2}$ | $85.35_{\pm5.5}$ | $75.00_{\pm14.8}$ | $90.02_{\pm6.8}$ | $93.50_{\pm4.9}$ | $57.70_{\pm7.1}$ | $45.92_{\pm8.5}$ | $56.14_{\pm7.2}$ | $93.56_{\pm5.5}$ | $94.76_{\pm4.4}$ | **5.00** |
| WPFS | $83.86_{\pm9.1}$ | $66.89_{\pm6.2}$ | $96.42_{\pm4.2}$ | $79.14_{\pm4.5}$ | $73.83_{\pm16.5}$ | $89.15_{\pm6.7}$ | $88.29_{\pm5.3}$ | $59.54_{\pm6.9}$ | $55.91_{\pm8.6}$ | $59.05_{\pm8.6}$ | $95.96_{\pm4.1}$ | $94.83_{\pm4.2}$ | **3.42** |
| TabNet | $64.54_{\pm12.9}$ | $61.16_{\pm9.2}$ | $71.64_{\pm17.7}$ | $50.87_{\pm13.8}$ | $50.00_{\pm16.9}$ | $65.75_{\pm17.7}$ | $41.38_{\pm9.6}$ | $49.08_{\pm9.3}$ | $39.57_{\pm11.6}$ | $53.19_{\pm9.4}$ | $81.27_{\pm9.7}$ | $75.11_{\pm10.2}$ | 13.58 |
| TabTransformer | $78.82_{\pm14.1}$ | $64.00_{\pm9.2}$ | $88.38_{\pm8.6}$ | $76.81_{\pm6.8}$ | $63.50_{\pm15.6}$ | $85.96_{\pm11.5}$ | $87.67_{\pm6.1}$ | $56.91_{\pm5.6}$ | $40.70_{\pm6.9}$ | $52.49_{\pm9.0}$ | $93.82_{\pm4.7}$ | $94.03_{\pm4.7}$ | 8.83 |
| CAE | $74.18_{\pm11.7}$ | $59.96_{\pm11.0}$ | $89.80_{\pm9.2}$ | $71.94_{\pm13.4}$ | $67.83_{\pm17.6}$ | $87.60_{\pm7.8}$ | $60.36_{\pm11.3}$ | $59.54_{\pm8.3}$ | $40.69_{\pm7.4}$ | $57.35_{\pm9.4}$ | $95.78_{\pm3.6}$ | $85.00_{\pm5.0}$ | 9.17 |
| LassoNet | $53.91_{\pm10.9}$ | $51.04_{\pm8.6}$ | $50.80_{\pm12.9}$ | $30.63_{\pm8.7}$ | $29.17_{\pm11.8}$ | $54.78_{\pm10.6}$ | $26.67_{\pm8.7}$ | $46.08_{\pm9.2}$ | $33.49_{\pm7.5}$ | $48.88_{\pm5.7}$ | $48.41_{\pm10.8}$ | $25.11_{\pm9.8}$ | 15.00 |
| MLP | $77.72_{\pm15.3}$ | $64.42_{\pm8.4}$ | $91.30_{\pm6.7}$ | $78.30_{\pm9.0}$ | $73.00_{\pm14.9}$ | $88.76_{\pm5.5}$ | $93.21_{\pm6.1}$ | $56.28_{\pm6.7}$ | $48.19_{\pm7.8}$ | $59.56_{\pm5.5}$ | $94.31_{\pm5.4}$ | $94.20_{\pm4.9}$ | 6.00 |
| Random Forest | $81.15_{\pm8.5}$ | $69.84_{\pm4.6}$ | $96.80_{\pm5.6}$ | $76.44_{\pm10.1}$ | $74.17_{\pm10.6}$ | $90.35_{\pm8.2}$ | $80.99_{\pm4.5}$ | $66.04_{\pm5.2}$ | $47.12_{\pm7.0}$ | $52.98_{\pm5.4}$ | $89.39_{\pm7.2}$ | $88.14_{\pm5.2}$ | 6.42 |
| LightGBM | $80.79_{\pm7.6}$ | $70.07_{\pm5.8}$ | $95.36_{\pm5.2}$ | $74.22_{\pm14.6}$ | $75.50_{\pm11.9}$ | $91.91_{\pm4.8}$ | $81.26_{\pm4.2}$ | $59.46_{\pm5.8}$ | $44.99_{\pm9.3}$ | $57.69_{\pm8.6}$ | $93.42_{\pm7.2}$ | $89.79_{\pm4.4}$ | 6.08 |
| GCN | $84.09_{\pm9.4}$ | $65.63_{\pm8.0}$ | $80.83_{\pm10.8}$ | $72.00_{\pm8.4}$ | $66.23_{\pm14.4}$ | $82.60_{\pm12.5}$ | $76.13_{\pm7.0}$ | $58.31_{\pm5.8}$ | $51.01_{\pm8.2}$ | $58.29_{\pm7.4}$ | $91.13_{\pm8.7}$ | $93.30_{\pm4.6}$ | 7.75 |
| GATv2 | $73.57_{\pm12.4}$ | $66.06_{\pm8.2}$ | $71.36_{\pm11.6}$ | $57.74_{\pm14.1}$ | $57.67_{\pm15.1}$ | $83.23_{\pm10.6}$ | $76.65_{\pm11.2}$ | $53.60_{\pm6.9}$ | $45.45_{\pm9.3}$ | $54.71_{\pm7.1}$ | $86.96_{\pm8.2}$ | $93.33_{\pm6.2}$ | 10.92 |
| **GCondNet (KNN)** | $85.02_{\pm9.0}$ | $65.92_{\pm8.7}$ | $96.18_{\pm4.9}$ | $80.70_{\pm5.5}$ | $76.67_{\pm12.9}$ | $90.38_{\pm5.6}$ | $94.33_{\pm4.1}$ | $58.62_{\pm7.0}$ | $51.70_{\pm8.8}$ | $59.34_{\pm8.9}$ | $95.96_{\pm4.2}$ | $95.20_{\pm3.8}$ | **1.92** |
| **GCondNet (SRD)** | $86.36_{\pm8.0}$ | $68.08_{\pm7.3}$ | $97.56_{\pm4.1}$ | $79.92_{\pm6.2}$ | $77.67_{\pm10.5}$ | $89.33_{\pm7.6}$ | $95.25_{\pm4.5}$ | $56.36_{\pm9.4}$ | $50.82_{\pm9.5}$ | $58.24_{\pm6.4}$ | $96.13_{\pm4.0}$ | $96.64_{\pm3.1}$ | |

Welling, 2017) and Graph Attention Network v2 (GATv2) (Brody et al., 2022). To ensure fairness, we employ a similar setup to GCondNet, constructing a KNN-based graph ($k = 5$) in which each node represents a sample connected to its five nearest samples based on cosine similarity, which is well-suited for high-dimensional data. Both GCN and GATv2 are trained in a transductive setting, incorporating test sample edges during training while masking nodes to prevent data leakage.

## 3.1 Overall Classification Performance

Our experiments in Table 1 show that GCondNet outperforms all 14 benchmark models on average, achieving a better overall rank across 12 real-world datasets – suggesting its effectiveness across diverse datasets. GCondNet is followed by WPFS, a specialised method for small sample-size and high-dimensional datasets, although GCondNet consistently outperforms it on 9 out of 12 tasks, providing improvements of up to 7%. Standard methods like LightGBM and Random Forest are competitive and perform well on some datasets, but their relative performance is sometimes highly dataset-dependent. For instance, in the case of the "tcga-survival" dataset, GCondNet (with an MLP backbone) enhances MLP's performance by 2%, but it still lags behind top methods like Random Forest, WPFS, and CAE, which exhibit an additional 1-7% better performance. This suggests that the additional feature selection capabilities of these approaches can lead to further improvements. While in this work we did not analyse this behaviour in detail, GCondNet, in principle, can readily handle this.

GCondNet consistently outperforms an MLP with the same architecture, showing substantial improvements. The most notable increases of 3-8% are observed in the five most extreme datasets, which have the smallest $N/D$ ratios. On those extreme datasets, GCondNet also improves stability compared to the standalone MLP, reducing the average standard deviation by over 3.5% when using the SRD version and 2.5% when using the KNN version. As the $N/D$ ratio increases, GCondNet continues to deliver superior performance to the baseline MLP, albeit with similar stability. These results underscore the role of GCondNet's inductive bias to mitigating overfitting and enhancing stability, particularly in datasets with extreme $N/D$ ratios.

We compare against GNNs on tabular data where relationships between samples are not explicitly provided. Despite the advantage of GCN and GATv2 of being trained in a transductive setting, GCondNet outperforms both methods across tasks. The performance gap ranges between 19-25% on three tasks and more than 5% on four other tasks. This indicates that models heavily reliant on *latent* structure present in tabular data, such as GNNs, are particularly sensitive to misspecifications during model construction. In contrast, GCondNet demonstrates resilience against such misspecifications, which we analyse in the following section.

The results also show that GCondNet outperforms other methods specialised for this data scenario by a large margin, such as DietNetworks, FsNet, SPINN, DNP, and more complex neural architectures for tabular data such as TabNet, TabTransformer, CAE and LassoNet. This finding aligns with recent research (Kadra et al., 2021), suggesting that well-regularised MLPs are more effective at handling tabular data than intricate architectures. Because our MLP baseline was already well-regularised, we attribute GCondNet's performance improvement to its inductive bias, which serves as an additional regularisation effect, as we further investigate in the next section.

Finally, we find that GCondNet consistently performs well with both KNN and SRD graphs, ranking high across various datasets. However, no clear distinction emerges between the two graph construction methods, and in the next section, we further analyse GCondNet's robustness to this choice.

## 3.2   Analysing the Inductive Bias of GCondNet

Having found that GCondNet excels on small-size and high-dimensional tasks, we analyse its inductive bias and robustness to different construction methods.

**GCondNet outperforms other initialisation schemes that do not use GNNs.** To understand the effect of leveraging the latent relationships between samples to parameterise neural networks (as GCondNet does), we investigate if other weight initialisation methods can imbue a similar inductive bias. To the best of our knowledge, all such existing methods necessitate external knowledge, like in (Li & Li, 2008; Ruiz et al., 2023). Consequently, *we propose* three novel weight initialisation schemes incorporating a similar inductive bias as GCondNet, making similar features having similar weights. These schemes generate feature embeddings $e^{(i)}$, which are then utilised to initialise the MLP's first layer $W_{\text{MLP}}^{[1]} = [e^{(1)}, e^{(2)}, ..., e^{(D)}]$. The feature embeddings are computed using Non-negative matrix factorisation (NMF), Principal Component Analysis (PCA), and Weisfeiler-Lehman (WL) algorithm (Weisfeiler & Leman, 1968). The latter is a parameter-free method to compute graph embeddings, often used to check whether two graphs are isomorphic. We apply the WL algorithm to the same SRD graphs as used for GCondNet and use the $j$-th graph embedding as the feature embedding $e_{\text{WL}}^{(j)}$. See Appendix F.3 for more details on these initialisation methods. We follow (Grinsztajn et al., 2022) and compute the normalised test balanced accuracy across all 12 datasets and 25 runs. We include the absolute accuracy numbers in Appendix F.3.

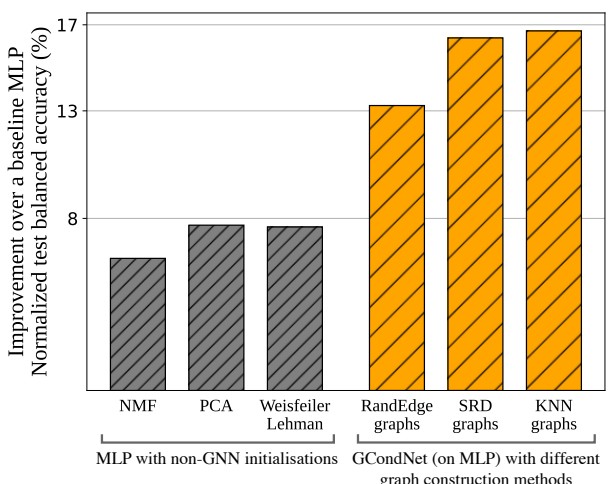

Figure 2: **The inductive bias of GCondNet robustly improves performance and cannot be replicated without GNNs.** We compute the normalised test balanced accuracy across all 12 datasets and 25 runs and report the relative improvement over a baseline MLP. First, we find that GCondNet is robust across various graph construction methods and provides consistent improvement over an equivalent MLP. Second, to assess the usefulness of the GNNs, we propose three weight initialisation methods designed to emulate GCondNet's inductive biases but without employing GNNs. The results show that GCondNet outperforms such methods, highlighting the effectiveness of the GNN-extracted latent structure.

Figure 2 shows that the specialised initialisation schemes (NMF, PCA, WL) outperform a standard MLP, and we observe that GCondNet with both SRD and KNN graphs further improves over these initialisations. These results suggest that initialisation methods that incorporate appropriate inductive biases, as in GCondNet, can outperform popular initialisation methods, which assume the weights should be independent at initialisation (Glorot & Bengio, 2010; He et al., 2015), which can lead to overfitting on small datasets.

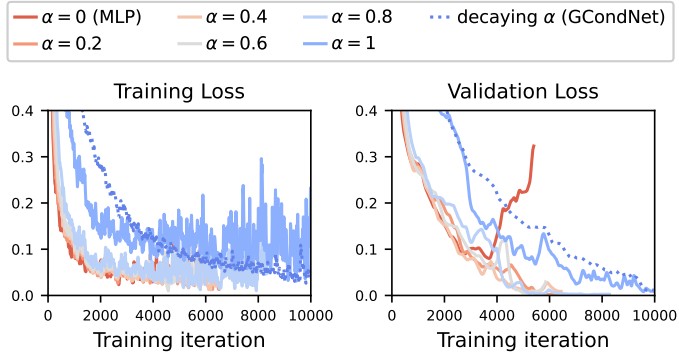

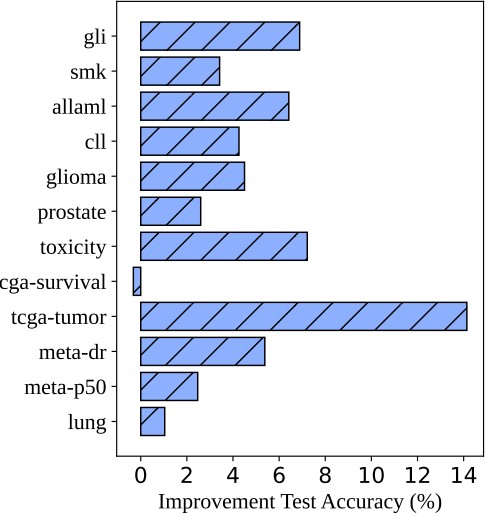

Figure 3: **GCondNet reduces overfitting.** The impact of varying the mixing coefficient $\alpha$ is illustrated through the training and validation loss curves (averaged over 25 runs) on 'toxicity'. We train GCondNet with linearly *decaying* $\alpha$, along with modified versions with *fixed* $\alpha$. Two observations are notable: (i) GCondNet exhibits less overfitting (evident from the converging validation loss) compared to an MLP ($\alpha = 0$), which overfits at the 4,000[th] iteration; (ii) decaying $\alpha$ enhances the training stability while improving the test-time accuracy by at least 2%.

Figure 4: GCondNet is versatile and can enhance various models beyond MLPs. When applied to TabTransformer, GCondNet consistently improves performance by up to 14%.

The advantage of using GNNs becomes more evident when comparing the performance of GCondNet, which incorporates a learnable GNN, to the Weisfeiler-Lehman method, which is a parameter-free method. Although both methods are applied on the same SRD graphs, GCondNet's higher performance by 7% underscores the crucial role of *training* GNNs to distil structure from the sample-wise relationships, exceeding the capabilities of other methods.

**GCondNet is robust to different graph construction methods.** As GCondNet is general and does not rely on knowing the ground-truth relationship between samples, we analyse its robustness to the user-defined method for defining such relationships. In addition to the KNN and SRD graphs from Section 2.1, we also consider graphs with random edges between samples (called "RandEdge" graphs). Specifically, we generate the RandEdge graphs with similar graph statistics to the SRD graphs but random graph edges (full details for creating RandEdge graphs are in Appendix F.3). We train GCondNet on 25 different data splits, and for each split, we sample RandEdge graphs five times – resulting in 125 trained models on RandEdge graphs.

We analyse two distinct facets of robustness. First, the *relative performance* compared to other benchmark models. We find that, generally, GCondNet, when using any of the KNN, SRD or RandEdge graphs, outperforms all 14 benchmark baselines from Table 1 and achieves a higher average rank. This suggests that GCondNet is robust to different graph-creation methods and maintains stable relative performance compared to other benchmark methods, even when the graphs are possibly misspecified.

Next, we analyse the *absolute performance* denoted by the numerical value of test accuracy. As expected, we find that RandEdge graphs are suboptimal (see Figure 2) and GCondNet performs better with more principled KNN or SRD graphs, which define similarity-based edges between samples. Nonetheless, the three graphs exhibit similar absolute performance with statistical significance (see Appendix F.3), making GCondNet resilient to misspecifications during graph construction. Moreover, the optimal graph construction method is task-dependent: SRD excels in seven datasets, KNN in another four, and RandEdge graphs in one. For instance, even with identical input data $X$ (as in 'tcga-survival' or 'tcga-tumor') – and thus identical graphs – the optimal graph construction method can differ based on the prediction task. We believe a limitation of this work is the lack of an optimal graph construction method. However, having an optimal graph construction method is non-trivial, as (i) different tasks rely on exploiting different modeling assumptions and structures in the data; and (ii) the GNN's oversmoothing issue (Chen et al., 2020) can

lead to computing just an "average" of the feature values. Lastly, we highlight that GCondNet is robust to the number of steps $n_\alpha$, as supported by the statistical tests in Appendix F.1.

**GCondNet's inductive bias serves as a regularisation mechanism.** To isolate GCondNet's inductive bias – which ensures that similar features have similar weights at *the beginning* of training – we train two versions of GCondNet: (i) with *decaying $\alpha$*, and (ii) a modified version with a *fixed* mixing coefficient $\alpha \in \{0, 0.2, 0.4, 0.6, 0.8, 1\}$ throughout the training process. As $\alpha \to 0$, the model becomes equivalent to an MLP, and as $\alpha \to 1$, the first layer is conditioned on the GNN-extracted structure.

We find that incorporating structure into the model serves as a regularisation mechanism and can help prevent overfitting. Figure 3 shows the training and validation loss curves for different $\alpha$ values. An MLP (equivalent to $\alpha = 0$) begins overfitting at around the 4,000[th] iteration, as shown by the inflexion point in the validation loss. In contrast, all models incorporating the structure between samples (i.e., $\alpha > 0$) avoid this issue and attain better validation loss.

The optimisation perspective further motivates decaying $\alpha$ rather than using a fixed value. Firstly, using a fixed $\alpha$ during training can lead to instabilities, such as high variance in the training loss. Secondly, a fixed $\alpha$ results in a test-time performance drop compared to the decaying version. In the case of "toxicity" (presented in Figure 3) this leads to at least 2% loss in performance compared to the decaying variant with 95.25%, ranging between $91.33\% - 93.08\%$ (with no particular trend) for different values of $\alpha \in \{0, 0.2, 0.4, 0.6, 0.8\}$. Training with a fixed $\alpha = 1$ results in a lower accuracy of 84.24%. We posit this occurs because the model is overly constrained on potentially incorrect graphs without sufficient learning capacity for other weights. The model gains flexibility by decaying $\alpha$ from $1 \to 0$, achieving better generalisation and increased stability.

**Extension to Transformers.** We highlight that GCondNet is a general framework for injecting graph-regularisation into various types of neural networks, and it can readily be applied to other architectures beyond an MLP. As a proof-of-principle, we apply GCondNet to TabTransformer (Huang et al., 2020), and the results in Figure 4 show consistent performance improvements by up to 14% (averaged over 25 runs). This highlights that GCondNet is general and can be applied to various downstream models.

**GCondNet is effective across various sample-to-feature ratios $N/D$.** We compare the performance of GCondNet (with an MLP backbone and KNN graphs) to an identical MLP baseline. Both models were trained under the same conditions. We selected four datasets ("allaml", "cll", "gli", "glioma") to cover a broad range of $N/D$ ratios, keeping the per-dataset number of samples $N$ constant. We adjust the number of features $D$ accordingly—for instance, a dataset with $N = 100$ samples had $D$ reduced from 10,000 to 20. We split the feature sets by sequentially removing features, repeating the process five times, and training the MLP and GCondNet with identical features. Full details of the setup and numerical results are in Appendix G.2.

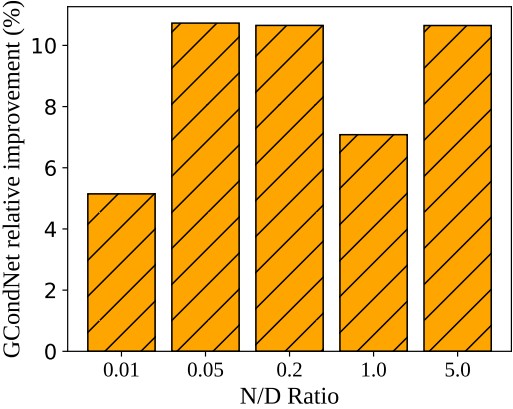

Figure 5: Improvement of GCondNet (with an MLP backbone) over an equivalent MLP baseline. This figure shows the relative increase in test balanced accuracy of GCondNet compared to the MLP baseline, averaged across datasets. GCondNet consistently enhances performance across various $N/D$ ratios, demonstrating its advantages even on small-dimensional data.

Figure 5 shows that GCondNet outperforms a baseline MLP by up to 10% (in relative terms) across various sample-to-feature ratios $N/D$. This indicates that GCondNet can also be effective for small-dimensional data. Moreover, GCondNet demonstrates greater stability and lower variance in performance, even with reduced feature sets.

## 4 Related Work

**"Diet" networks:** Our focus is learning problems on high-dimensional tabular datasets with a limited number of samples. As such, our method takes inspiration from "diet" methods such as DietNetworks (Romero et al., 2017), FsNet (Singh & Yamada, 2023) and WPFS (Margeloiu et al., 2023), which rely on auxiliary networks to predict (and in turn reduce) the number of learnable parameters in the first layer of an underlying feed-forward network. However, GCondNet differs from "diet" methods in two important ways: (i) "diet" methods require a well-defined strategy for computing feature embeddings, and their performance is highly sensitive to this choice. In contrast, GCondNet defines graphs between samples and uses a GNN to learn the feature embeddings; (ii) GCondNet provides a different inductive bias which leverages the implicit relationships between samples (via the learned graph embeddings $\boldsymbol{w}^{(i)}$).

Out of all "diet" methods, GCondNet is most closely related to PLATO (Ruiz et al., 2023), as both methods employ GNNs as auxiliary networks to parameterise the predictor network. However, the similarities end there, and we highlight two key differences: (i) PLATO relies on domain knowledge, making it inapplicable when such information is unavailable, which is common. In contrast, GCondNet is more general, as it can be applied to any tabular dataset without requiring domain knowledge but can still utilise it when available. (ii) PLATO constructs a single graph between features, whereas GCondNet creates multiple graphs between samples. This distinction is crucial, as PLATO leverages the relationships among features, while our method focuses on leveraging the relationships between samples (in addition to the relationships between features learnt by the MLP predictor itself).

**Graph-based approaches for tabular data**, including semi-supervised approaches, construct a graph between samples to capture the underlying relationships. The graphs are created using either a user-defined metric or by learning a latent graph between samples. Recent methods apply GNNs to these graphs, and our work distinguishes itself from such tabular data approaches in three ways.

1. We use GNNs indirectly and *only during training* to improve an underlying MLP predictor. Once trained, we store the MLP predictor's final weights, eliminating the need for GNNs during inference. Test input samples are subsequently processed exclusively through the predictor MLP. In contrast, GNN approaches to tabular data (You et al., 2020; Wu et al., 2021; Du et al., 2022; Fatemi et al., 2021; Satorras & Bruna, 2018) directly employ GNNs *for inference* on new inputs, including making predictions (You et al., 2020; Du et al., 2022; Fatemi et al., 2021; Satorras & Bruna, 2018) and performing feature imputation (You et al., 2020; Wu et al., 2021).

2. Our graph structure is different. GCondNet generates many graphs between samples (one for each feature) and then extracts graph embeddings $\boldsymbol{w}^{(j)}$ to parameterise a predictor network. This approach is novel and clearly distinguishes it from other work such as (You et al., 2020; Wu et al., 2021; Du et al., 2022), which generate graphs connecting features and samples. Both (You et al., 2020; Wu et al., 2021) construct a bipartite graph between samples and features, while (Du et al., 2022) creates a hyper-graph where each sample is a node linked to corresponding feature nodes (specifically for discrete data).

3. Graph-based approaches for tabular data often introduce additional assumptions that may be suboptimal or inapplicable. For example, (Kazi et al., 2022; Zhou et al., 2022; Fatemi et al., 2021) create a graph between samples and rely on the *smoothness assumption*, which posits that neighbouring instances share the same labels. As demonstrated in Section 3.1, such assumptions can be suboptimal for high-dimensional data. Concerning *dataset assumptions*, (Satorras & Bruna, 2018) addresses few-shot learning, which requires a substantial meta-training set comprising similar tasks. In contrast, our approach focuses on learning from small datasets without assuming the presence of an external meta-training set. The work of (Fatemi et al., 2021) infers a latent graph, focusing on either images or tabular data with a maximum of 30 features. In comparison, our research explores tabular datasets containing up to 20000 features.

**Feature selection.** When faced with high-dimensional data, machine learning models are presented with increased degrees of freedom, making prediction tasks more challenging, especially on small sample-size tasks. To address this issue, various feature selection methods have been proposed to reduce the dimensionality of the data (Tibshirani, 1996; Feng & Simon, 2017; Liu et al., 2017; Singh & Yamada, 2023; Balın et al., 2019; Margeloiu et al., 2023; Lemhadri et al., 2021). All these methods aim to model the relationships between

features (i.e., determining which features are similar or irrelevant to the task), but they do not consider the relationships between samples. In contrast, GCondNet uses a GNN to extract the relationships between samples, while the MLP predictor learns the relationships between features.

**Neural networks for tabular data.** More broadly, our work is related to neural network methods for tabular data. Recent methods include various inductive biases, such as taking inspiration from tree-based methods (Katzir et al., 2020; Hazimeh et al., 2020; Popov et al., 2020; Yang et al., 2018), including attention-based modules (Arık & Pfister, 2021; Huang et al., 2020), or modelling multiplicative feature interactions (Qin et al., 2021). For a recent review on neural networks for tabular data, refer to (Borisov et al., 2022). However, these methods are generally designed for large sample size datasets, and their performance can vary on different datasets (Gorishniy et al., 2021), making them unsuitable for small-size and high-dimensional tasks. In contrast, our method is specifically designed for small-size high-dimensional tasks. Lastly, TabPFN (Hollmann et al., 2022) is a recent pre-trained Transformer using in-context learning for prediction, which can scale only up to 100 features, making it inapplicable for our high-dimensional datasets (of up to 22,000 features).

## 5 Conclusion

We introduce GCondNet, a general method to improve neural network predictors on small and high-dimensional tabular datasets. The key innovation of GCondNet lies in exploiting the "implicit relationships" between *samples* by performing "soft parameter-sharing" to constrain the model's parameters. We also propose *sample-wise multiplex graphs*, a novel and general approach to identify and use these potential relationships between samples by constructing many graphs between samples, one for each feature. We then use Graph Neural Networks (GNNs) to extract any implicit structure and condition the parameters of the first layer of an underlying predictor network. Unlike other methods, which require external application-specific knowledge graphs, our method is general and can be applied to any tabular dataset.

We evaluate 12 classification tasks on biomedical datasets – in real applications, this could mean identifying biomarkers for different diseases – and show that GCondNet outperforms 14 benchmark methods and is robust to different graph construction methods. We also show that the GNN-extracted structure serves as a regularisation mechanism for reducing overfitting. Future work can investigate using the learned structures to obtain insights into the dataset, such as detecting mislabeled data points or outliers.

### Broader Impact Statement

This paper presents a novel method that aims to advance the field of machine learning by offering a new direction for leveraging the implicit sample relationships in machine learning, which is particularly beneficial for data-scarce tasks. This work can also serve as a basis for more interpretable approaches using the learned data structures. For critical domains such as medicine, GCondNet can provide patient/cohort-wise insights through post-hoc mechanisms such as graph concept-based explanations (Magister et al., 2021; 2022). From a machine learning perspective, GCondNet may also provide valuable insights into the dataset, such as identifying difficult training samples in the context of curriculum learning (Bengio et al., 2009).

Our work's impact is to advance machine learning capabilities in critical fields such as medicine and scientific research, particularly in contexts where data availability is limited. By improving model performance in settings with scarce data, our approach supports essential research in early-phase clinical trials (Weissler et al., 2021; Zame et al., 2020) – where typically only a small number of patients are enrolled – and it can help in identifying subtle patterns and relationships from small datasets. By handling complex, high-dimensional datasets, GCondNet can benefit genomics research, enabling better analysis of genetic variations and interactions with limited experimental data, thus supporting the discovery of genetic markers and pathways (Alharbi & Vakanski, 2023; Way & Greene, 2019). We do not foresee harmful applications for our method.

### Acknowledgements

The authors thank Mateo Espinosa Zarlenga and Ramon Viñas Torné for their feedback on earlier versions of the manuscript. We also thank Iulia Duta and Pietro Barbiero for their early discussions on graph neural networks. NS acknowledges the support of the U.S. Army Medical Research and Development Command

of the Department of Defense; through the FY22 Breast Cancer Research Program of the Congressionally Directed Medical Research Programs, Clinical Research Extension Award GRANT13769713. Opinions, interpretations, conclusions, and recommendations are those of the authors and are not necessarily endorsed by the Department of Defense.

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

# Appendix: GCondNet

## Table of Contents

# A  GCondNet Training Pseudocode

---

**Algorithm 2** Training GCondNet

---

**Input**: training data $\boldsymbol{X} \in \mathbb{R}^{N \times D}$, training labels $\boldsymbol{y} \in \mathbb{R}^N$, classification network $f_{\theta_{\mathrm{MLP}}}$, graph neural network $g_{\theta_{\mathrm{GNN}}}$, node aggregation function $f_{\mathrm{agg}}$, graph creation method $h(\cdot)$, steps for linear decay $n_\alpha$

1:  **for** each feature $j = 1, 2, ..., D$ **do**
2:      $\mathcal{G}_j = h(\boldsymbol{X}_{:,j})$          ▷ Generate the sample-wise multiplex graphs
3:  **end for**
4:  $\boldsymbol{W}_{\mathrm{scratch}} = 0$          ▷ Initialise auxiliary weight matrix
5:  **for** each mini-batch $B = \{(\boldsymbol{x}^{(i)}, y_i)\}_{i=1}^b$ **do**
6:      **for** each feature $j = 1, 2, ..., D$ **do**
7:          node-embeddings $= g_{\theta_{\mathrm{GNN}}}(\mathcal{G}_j)$
8:          $\boldsymbol{w}^{(j)} = f_{\mathrm{agg}}(\text{node-embeddings})$ ▷ Aggregate all node embeddings to obtain the graph embedding $\boldsymbol{w}^{(j)} \in \mathbb{R}^K$
9:      **end for**
10:     $\boldsymbol{W}_{\mathrm{GNN}} = [\boldsymbol{w}^{(1)}, \boldsymbol{w}^{(2)}, ..., \boldsymbol{w}^{(D)}]$          ▷ Horizontally concatenate the graph embeddings

11:     $\alpha = \max(0, 1 - (i/n_\alpha))$          ▷ Compute mixing coefficient
12:     $\boldsymbol{W}_{\mathrm{MLP}}^{[1]} \leftarrow \alpha \boldsymbol{W}_{\mathrm{GNN}} + (1 - \alpha)\boldsymbol{W}_{\mathrm{scratch}}$          ▷ Compute the weight matrix of the first layer
13:     Make $\boldsymbol{W}_{\mathrm{MLP}}^{[1]}$ the weight matrix of the first layer of $f_{\theta_{\mathrm{MLP}}}$

14:     **for** each sample $i = 1, 2, ..., b$ **do**
15:         $\hat{y}_i = f_{\theta_{\mathrm{MLP}}}(\boldsymbol{x}^{(i)})$
16:     **end for**
17:     $\hat{\boldsymbol{y}} \leftarrow [\hat{y}_1, \hat{y}_2, ..., \hat{y}_b]$          ▷ Concatenate all predictions
18:     Compute training loss $L = \mathrm{CrossEntropyLoss}(\boldsymbol{y}, \hat{\boldsymbol{y}})$
19:     Compute the gradient of the loss $L$ w.r.t.
20:         $\theta_{\mathrm{MLP}}, \theta_{\mathrm{GNN}}, \boldsymbol{W}_{\mathrm{scratch}}$ using backpropagation
21:     Update the parameters:
22:         $\theta_{\mathrm{MLP}} \leftarrow \theta_{\mathrm{MLP}} - \nabla_{\theta_{\mathrm{MLP}}} L$
23:         $\theta_{\mathrm{GNN}} \leftarrow \theta_{\mathrm{GNN}} - \nabla_{\theta_{\mathrm{GNN}}} L$
24:         $\boldsymbol{W}_{\mathrm{scratch}} \leftarrow \boldsymbol{W}_{\mathrm{scratch}} - \nabla_{\boldsymbol{W}_{\mathrm{scratch}}} L$
25:  **end for**
**Return**: Trained models $f_{\theta_{\mathrm{MLP}}}, g_{\theta_{\mathrm{GNN}}}$ and $\boldsymbol{W}_{\mathrm{MLP}}^{[1]}$

---

# B  Sparse Relative Distance (SRD) Graph Construction

We propose a novel similarity-based method, Sparse Relative Distance (SRD), for creating edges between node (representing samples) in a graph. It assumes that similar samples should be connected and use this principle to create edges $\mathcal{E}_j$ in the $j^{\mathrm{th}}$ graph. The method also includes an accept-reject step to sparsify the graph. Specifically, SRD work as follows:

1. For each node $i$ create a set of candidate edges $\mathbb{C}_i$ by identifying all samples $l$ with a feature value within a certain distance, $dist$, of the corresponding feature value of sample $i$. Specifically, we include all samples $l$ such that $|\boldsymbol{X}_{i,j} - \boldsymbol{X}_{l,j}| \leq dist$, where $dist$ is defined as 5% of the absolute difference between the $5^{\mathrm{th}}$ and $95^{\mathrm{th}}$ percentiles of all values of feature $j$ (to eliminate the effect of outlier feature values).

2. Perform a Bernoulli trial with probability $size(\mathbb{C}_i)/N_{\mathrm{train}}$ for each node $i$. If the trial outcome is positive, create undirected edges between node $i$ and all nodes within the candidate set $\mathbb{C}_i$. If the outcome is negative, no new edges are created. This sampling procedure results in sparser graphs, which helps alleviate the issue of oversmoothing (Chen et al., 2020) commonly encountered in GNNs. Oversmoothing occurs when the model produces similar embeddings for all nodes in the graph, effectively 'smoothing out' any differences in the graph structure. It is worth noting that a node can also acquire new edges as part of the candidate set of other nodes.

   This process results in a network topology in which nodes with larger candidate sets are more likely to have more connections. Intuitively, this means that 'representative' samples become the centres of node clusters in the network.

3. To further prevent oversmoothing, if a node has more than 25 connections, we randomly prune some of its edges until it has exactly 25 connections. This is because samples with highly frequent values can have an excessive number of connections, which can result in oversmoothing.

Table B.1: Key statistics of the graphs created using the Sparse Relative Distance (SRD) with $dist = 5\%$.

| Dataset | Node degree | Edges of full graph (%) |
|---|---|---|
| cll | $3.88 \pm 6.81$ | 9.96 |
| lung | $5.16 \pm 9.46$ | 7.37 |
| meta-p50 | $3.97 \pm 6.73$ | 5.56 |
| meta-dr | $3.92 \pm 6.69$ | 5.48 |
| prostate | $11.28 \pm 17.96$ | 31.77 |
| smk | $3.94 \pm 6.61$ | 5.93 |
| tcga-survival | $7.7 \pm 12.03$ | 10.77 |
| tcga-tumor | $7.45 \pm 11.84$ | 10.42 |
| toxicity | $3.1 \pm 5.42$ | 5.12 |

## C Datasets

Table C.2: Details of the 12 real-world biomedical datasets used for experiments. The datasets contain between 72-200 samples, and the number of features is $17 - 262$ times larger than the number of samples.

| Dataset | # samples (N) | # features (D) | D/N | # classes | # samples per class |
|---|---|---|---|---|---|
| allaml | 72 | 7129 | 99 | 2 | 25, 47 |
| cll | 111 | 11340 | 102 | 3 | 11, 49, 51 |
| gli | 85 | 22283 | 262 | 2 | 26, 59 |
| glioma | 50 | 4434 | 89 | 4 | 7, 14, 14, 15 |
| lung | 197 | 3312 | 17 | 4 | 17, 20, 21, 139 |
| meta-dr | 200 | 4160 | 21 | 2 | 61, 139 |
| meta-p50 | 200 | 4160 | 21 | 2 | 33, 167 |
| prostate | 102 | 5966 | 58 | 2 | 50, 52 |
| smk | 187 | 19993 | 107 | 2 | 90, 97 |
| tcga-survival | 200 | 4381 | 22 | 2 | 78, 122 |
| tcga-tumor | 200 | 4381 | 22 | 3 | 25, 51, 124 |
| toxicity | 171 | 5748 | 34 | 4 | 39, 42, 45, 45 |

All datasets are publicly available and summarised in Table C.2. Eight datasets are open-source (Li et al., 2018) and available `https://jundongl.github.io/scikit-feature/datasets.html`online: **CLL-SUB-111** (called 'cll') (Haslinger et al., 2004), **GLI_85** (called 'gli') (Freije et al., 2004), **lung** (Bhattacharjee et al., 2001), **Prostate_GE** (called 'prostate') (Singh et al., 2002), **SMK-CAN-187** (called 'smk') (Spira et al., 2007), **TOX-171** (called 'toxicity') (Bajwa et al., 2016), as well as **allaml** and **glioma**, for which the original reference was not available.

We created four additional datasets following the methodology presented in Margeloiu et al. (2023):

- Two datasets from the **METABRIC** (Curtis et al., 2012) dataset. We combined the molecular data with the clinical label 'DR' to create the **'meta-dr'** dataset, and we combined the molecular data with the clinical label 'Pam50Subtype' to create the **'meta-p50'** dataset. Because the label 'Pam50Subtype' was very imbalanced, we transformed the task into a binary task of basal vs non-basal by combining the classes 'LumA', 'LumB', 'Her2', 'Normal' into one class and using the remaining class 'Basal' as the second class. For both 'meta-dr' and 'meta-p50' we selected the Hallmark gene set (Liberzon et al., 2015) associated with breast cancer, and the new datasets contain 4160 expressions (features) for each patient. We created the final datasets by randomly sampling 200 patients stratified because we are interested in studying datasets with a small sample-size.

- Two datasets from the **TCGA** (Tomczak et al., 2015) dataset. We combined the molecular data and the label 'X2yr.RF.Surv' to create the **'tcga-survival'** dataset, and we combined the molecular data and the label 'tumor_grade' to create the **'tcga-tumor'** dataset. For both 'tcga-survival' and 'tcga-tumor' we selected the Hallmark gene set (Liberzon et al., 2015) associated with breast cancer, leaving 4381 expressions (features) for each patient. We created the final datasets by randomly sampling 200 patients stratified because we are interested in studying small sample-size datasets.

**Dataset processing.** Before training the models, we apply Z-score normalisation to each dataset. Specifically, on the training split, we learn a simple transformation to make each column of $X_{train} \in \mathbb{R}^{N_{train} \times D}$ have zero mean and unit variance. We apply this transformation to the validation and test splits during cross-validation.

## D  Reproducibility: Benchmarks methods, Training Details and Hyper-parameters

**Software implementation.** We implemented `GCondNet` using PyTorch 1.12 (Paszke et al., 2019), an open-source deep learning library with a BSD licence. We implemented the GNN within `GCondNet`, and the GCN and GATv2 benchmarks using PyTorch-Geometric (Fey & Lenssen, 2019), an open-source library for implementing Graph Neural Networks with an MIT licence. We train using a library `https://github.com/Lightning-AI/lightning` Pytorch-ligthning built on top of PyTorch and released under an Apache Licence 2.0. All numerical plots and graphics have been generated using Matplotlib 3.6, a Python-based plotting library with a BSD licence. The model architecture Figure 1 was generated using `https://github.com/jgraph/drawio` draw.io, a free drawing software under Apache License 2.0.

As for the other benchmarks, we implement MLP, CAE and DietNetworks using PyTorch 1.12 (Paszke et al., 2019), Random Forest using scikit-learn (Pedregosa et al., 2011) (BSD license), LightGBM using the lightgbm library (Ke et al., 2017) (MIT licence) and TabNet (Arık & Pfister, 2021) using the `https://github.com/dreamquark-ai/tabnet`implementation (MIT licence) from Dreamquark AI. We use the MIT-licensed `https://github.com/andreimargeloiu/WPFS` implementation of WPFS made public by (Margeloiu et al., 2023). We re-implement FsNet (Singh & Yamada, 2023) in PyTorch 1.12 (Paszke et al., 2019) because the official code implementation contains differences from the paper, and they used a different evaluation setup from ours (they evaluate using unbalanced accuracy, while we run multiple data splits and evaluate using balanced accuracy). We use the `https://github.com/lasso-net/lassonet` official implementation of LassoNet (MIT licence), and the `https://github.com/jjfeng/spinn` official implementation of SPINN (no licence).

**Computing Resources.** All our experiments are run on a single machine from an internal cluster with a GPU Nvidia Quadro RTX 8000 with 48GB memory and an Intel(R) Xeon(R) Gold 5218 CPU with 16 cores (at 2.30GHz). The operating system was Ubuntu 20.4.4 LTS. We estimate that to carry out the full range of experiments, comprising both prototyping and initial experimental phases, we needed to train around 11000 distinct models, which we estimate required 1000 to 1200 GPU hours.

**GCondNet architecture and settings**. GCondNet is leveraging an MLP backbone model of three layers with $100, 100, 10$ neurons. After each linear layer we add LeakyReLU non-linearity with slope 0.01, batch normalisation (Ioffe & Szegedy, 2015) and dropout (Srivastava et al., 2014): we tune the dropout probability $p \in \{0.2, 0.4\}$ on the validation accuracy. The last layer has softmax activation. The layers following the first one are initialized using a standard Kaiming method (He et al., 2015), which considers the activation. The GNN within `GCondNet` is a Graph Convolutional Network (GCN) (Kipf & Welling, 2017) with two layers of size 200 and 100. After the first GCN layer, we use a ReLU non-linearity and dropout with $p = 0.5$. The permutation invariant function $f_{\mathrm{agg}}$ for computing graph embeddings is global average pooling.[3]

We intend GCondNet's computation overhead to be minimal, thus our GCondNet retains the same training hyper-parameters as the optimised backbone MLP model. We initially performed hyperparameter tuning on an MLP. After selecting the optimal hyperparameters, we retrained the GCondNet with the same MLP backbone and training settings, using two graph construction methods, resulting in two variants: GCondNet (KNN) and GCondNet (SRD), as shown in Table 1. Consequently, the time needed to tune GCondNet is

---

[3]Using hierarchical pooling methods (Ying et al., 2018; Ranjan et al., 2020) resulted in unstable training and significantly poor performance.

essentially equivalent to tuning the backbone model. Specifically, we train GCondNet for 10000 steps with a batch size of 8 and optimise using AdamW (Loshchilov & Hutter, 2019) with a fixed learning rate of $1e - 4$. We decay the mixing coefficient $\alpha$ over $n_\alpha = 200$ training steps, although we found that GCondNet is robust to the number of steps $n_\alpha$, as supported by the statistical tests in Appendix F.1. We use early stopping with patience 200 steps on the validation loss across all experiments.

**Training details for all benchmark methods.** Here, we present the training settings for all benchmark models, and we discuss hyper-parameter tuning in the next paragraph. We train using 5-fold cross-validation with 5 repeats (training 25 models each run). For each run, we select 10% of the training data for validation. We perform a fair comparison whenever possible: for instance, we train all models using a weighted loss (e.g., weighted cross-entropy loss for neural networks), evaluate using balanced accuracy, and use the same classification network architecture for GCondNet, MLP, WPFS, FsNet, CAE and DietNetworks.

- **WPFS, DietNetworks, CAE, FsNet** have three hidden layers of size $100, 100, 10$. The Weight Predictor Network and the Sparsity Network have four hidden layers $100, 100, 100, 100$. They are trained for 10,000 steps using early stopping with patience 200 steps on the validation cross-entropy and gradient clipping at 2.5. For **CAE** and **FsNet** we use the suggested annealing schedule for the concrete nodes: exponential annealing from temperature 10 to 0.01. On all datasets, **DietNetworks** performed best with not decoder. For **WPFS** we use the NMF embeddings suggested in Margeloiu et al., 2023.

- For **GCN**, we used two graph convolutional layers with 200 and 100 neurons, followed by a linear layer with softmax activation for class label computation. ReLU and dropout with $p = 0.5$ follow each convolutional layer. We train using AdamW (Loshchilov & Hutter, 2019), tune the learning rate in $[1e - 3, 3e - 3, 1e - 4]$, and select the best model on the validation accuracy.

- For **GATv2**, we used two graph attentional layers with dropout $p = 0.5$. The first attention layer used 4 attention heads of size 100, and the second used one head of size 400. ReLU and dropout with $p = 0.5$ follow each attention layer. A linear layer with softmax activation for class label computation follows the attention layers. We train using AdamW (Loshchilov & Hutter, 2019), tune the learning rate in $[1e - 3, 3e - 3, 1e - 4]$, and select the best model on the validation accuracy.

- **LassoNet** has three hidden layers of size $100, 100, 10$. We use dropout 0.2, and train using AdamW (with betas $0.9, 0.98$) and a batch size of 8. We train using a weighted loss. We perform early stopping on the validation set.

- For **Random Forest**, we used 500 estimators, feature bagging with the square root of the number of features, and used balanced weights associated with the classes.

- For **LightGBM** we used 200200 estimators, feature bagging with 30

- For **TabNet**, we use width 8 for the decision prediction layer and the attention embedding for each mask (larger values lead to severe overfitting) and 1.5 for the feature re-usage coefficient in the masks. We use three steps in the architecture, with two independent and two shared Gated Linear Units layers at each step. We train using Adam (Kingma & Ba, 2015) with momentum 0.3 and gradient clipping at 2.

- For **TabTransformer**, we use only the head for continuous features, as our datasets do not contain categorical features. For a fair comparison, we use the same architecture as the MLP following the initial layer normalization, and train the model with the optimal settings of the MLP.

- For **SPINN**, we followed the results from the ablations in the original paper and tuned the sparse group lasso hyper-parameter $\lambda \in \{0, 0.001, 0.0032, 0.1\}$, the group lasso hyper-parameter $\alpha \in \{0.9, 0.99, 0.999\}$ in the sparse group lasso, the ridge-param $\lambda_0 \in \{0, 0.0001\}$, and train for at most 1,000 steps.

- For **DNP** we did not find any suitable implementation, and used SPINN with different settings as a proxy for DNP (because DNP is a greedy approximation to optimizing the group lasso, and SPINN optimises directly a group lasso). Specifically, our proxy for DNP results is SPINN trained for at most 1000 iterations, with $\alpha = 1$ for the group lasso in the sparse group lasso, ridge-param $\lambda_0 = 0.0001$, and we tuned the sparse group lasso hyper-parameter $\lambda \in \{0, 0.001, 0.0032, 0.1\}$.

**Hyper-parameter tuning.** For each model, we use random search and previous experience to find a good range of hyper-parameter values that we can investigate in detail. We then performed a grid search and

ran 25 runs for each hyper-parameter configuration. We selected the best hyper-parameter based on the average validation accuracy across the 25 runs.

For the MLP and DietNetworks we individually grid-searched learning rate $\in \{0.003, 0.001, 0.0003, 0.0001\}$, batch size $\in \{8, 12, 16, 20, 24, 32\}$, dropout rate $\in \{0, 0.1, 0.2, 0.3, 0.4, 0.5\}$. We found that learning rate 0.003, batch size 8 and dropout rate 0.2 work well across datasets for both models, and we used them in the presented experiments. In addition, for DietNetworks we also tuned the reconstruction hyper-parameter $\lambda \in \{0, 0.01, 0.03, 0.1, 0.3, 1, 3, 10, 30\}$ and found that across dataset having $\lambda = 0$ performed best. For WPFS we used the best hyper-parameters for the MLP and tuned only the sparsity hyper-parameter $\lambda \in \{0, 3e-6, 3e-5, 3e-4, 3e-3, 1e-2\}$ and the size of the feature embedding $\in \{20, 50, 70\}$. For FsNet we grid-search the reconstruction parameter in $\lambda \in \{0, 0.2, 1, 5\}$ and learning rate in $\{0.001, 0.003\}$. For LightGBM we performed grid-search for the learning rate in $\{0.1, 0.01\}$ and maximum depth in $\{1, 2\}$. For Random Forest, we performed a grid search for the maximum depth in $\{3, 5, 7\}$ and the minimum number of samples in a leaf in $\{2, 3\}$. For TabNet we searched the learning rate in $\{0.01, 0.02, 0.03\}$ and the $\lambda$ sparsity hyper-parameter in $\{0.1, 0.01, 0.001, 0.0001\}$, as motivated by (Yang et al., 2022). For GCN and GATv2 we searched the learning rate in $\{1e-3, 3e-3, 1e-4\}$. We selected the best hyper-parameters (Table D.3) on the weighted cross-entropy (except Random Forest, for which we used weighted balanced accuracy).

Table D.3: Best performing hyper-parameters for each benchmark model across datasets.

| | Random Forest | | LightGBM | | TabNet | | GCN | GATv2 | FsNet | | CAE | WPFS | |
|---|---|---|---|---|---|---|---|---|---|---|---|---|---|
| | max depth | min samples leaf | learning rate | max depth | learning | $\lambda$ sparsity | learning rate | learning rate | learning rate | $\lambda$ reconstruction | annealing iterations | $\lambda$ sparsity | embedding size |
| allaml | 3 | 3 | 0.1 | 2 | 0.03 | 0.0001 | 0.001 | 0.003 | 0.003 | 0 | 1000 | 0 | 50 |
| cll | 3 | 3 | 0.1 | 2 | 0.03 | 0.001 | 0.003 | 0.003 | 0.003 | 0 | 1000 | $3e-4$ | 70 |
| gli | 3 | 3 | 0.1 | 1 | 0.03 | 0.1 | 0.001 | 0.003 | 0.003 | 0 | 300 | $3e-4$ | 50 |
| glioma | 3 | 3 | 0.01 | 2 | 0.03 | 0.001 | 0.003 | 0.003 | 0.003 | 0 | 1000 | $3e-3$ | 50 |
| lung | 3 | 2 | 0.1 | 1 | 0.02 | 0.1 | 0.003 | 0.001 | 0.001 | 0 | 1000 | $3e-5$ | 20 |
| meta-p50 | 7 | 2 | 0.01 | 2 | 0.02 | 0.001 | 0.003 | 0.003 | 0.003 | 0 | 1000 | $3e-6$ | 50 |
| meta-dr | 7 | 2 | 0.1 | 1 | 0.03 | 0.1 | 0.003 | 0.003 | 0.003 | 0 | 300 | 0 | 50 |
| prostate | 5 | 2 | 0.1 | 2 | 0.02 | 0.01 | 0.0001 | 0.0001 | 0.003 | 0 | 1000 | $3e-3$ | 50 |
| smk | 5 | 2 | 0.1 | 2 | 0.03 | 0.001 | 0.001 | 0.0001 | 0.003 | 0 | 1000 | $3e-5$ | 50 |
| tcga-survival | 3 | 3 | 0.1 | 1 | 0.02 | 0.01 | 0.003 | 0.003 | 0.003 | 0 | 300 | $3e-5$ | 50 |
| tcga-tumor | 3 | 3 | 0.1 | 1 | 0.02 | 0.01 | 0.003 | 0.001 | 0.003 | 0 | 300 | $3e-5$ | 50 |
| toxicity | 5 | 3 | 0.1 | 2 | 0.03 | 0.1 | 0.003 | 0.0001 | 0.001 | 0.2 | 1000 | $3e-5$ | 50 |

**LassoNet unstable training.** We used the official implementation of LassoNet (`https://github.com/lasso-net/lassonet`), and we successfully replicated some of the results in the LassoNet paper (Lemhadri et al., 2021). However, LassoNet was unstable on all datasets we trained on. In our experiments, we grid-searched the $L_1$ penalty coefficient $\lambda \in \{0.001, 0.01, 0.1, 1, 10, 'auto'\}$ and the hierarchy coefficient $M \in \{0.1, 1, 3, 10\}$. These values are suggested in the paper and used in the examples from the official codebase. For all hyper-parameter combinations, LassoNet's performance was equivalent to a random classifier (e.g., 25% balanced accuracy for a 4-class problem).

# E   Computational Cost

We present the time and number of steps required to train GCondNet. Note that we designed GCondNet to use the same training hyperparameters as the optimised backbone MLP model. Therefore, when implementing GCondNet, one should first perform hyperparameter tuning on the backbone model, which is quicker than training GCondNet. After selecting the optimal hyperparameters, retrain GCondNet using the same backbone and training settings, but choose a graph construction method (e.g., KNN or SRD). Consequently, the time needed to tune GCondNet is essentially the same as for the backbone model, with the only difference being the final one-off GCondNet training, for which we show the computational overhead here.

## E.1   Training time

Table E.4: Average training time (in minutes) of a model with optimal hyper-parameters. In general, GCondNet trains slower than the benchmarks: GCondNet takes 8.5 minutes to train across datasets, other competitive "diet" networks, such as WPFS, take 7.7 minutes, and an MLP takes 5.4 minutes.

| Dataset | GCondNet | MLP | WPFS | DietNetworks | FsNet |
|---|---|---|---|---|---|
| cll | 12 | 6.1 | 7.7 | 8.1 | 5.4 |
| lung | 11 | 6.1 | 9.9 | 10.6 | 6.1 |
| meta-p50 | 9.8 | 6.6 | 8.9 | 8.7 | 4.3 |
| meta-dr | 3.6 | 3 | 4.6 | 4.9 | 4.8 |
| prostate | 9.6 | 7.2 | 9.9 | 7.9 | 3.5 |
| smk | 9 | 6.2 | 8.9 | 5.9 | 4.3 |
| tcga-survival | 3.8 | 3 | 4.2 | 4.7 | 5.3 |
| tcga-tumor | 4 | 3.5 | 5 | 4.4 | 4.5 |
| toxicity | 13.7 | 7.3 | 10 | 8.2 | 5.9 |
| **Average minutes** | 8.5 | 5.4 | 7.7 | 7 | 4.9 |

## E.2   Number of steps for convergence

Table E.5: Number of steps for convergence.

| | MLP | GCondNet with different graphs | | |
|---|---|---|---|---|
| | | KNN graphs | SRD graphs | RandEdge graphs |
| cll | 1697 | 4020 | 4526 | 4298 |
| lung | 2836 | 6927 | 6777 | 6935 |
| meta-p50 | 2952 | 5504 | 5365 | 5242 |
| meta-dr | 1331 | 1657 | 1577 | 1598 |
| prostate | 2347 | 4307 | 4540 | 4535 |
| smk | 1379 | 2175 | 2102 | 2218 |
| tcga-survival | 1320 | 1541 | 1606 | 1588 |
| tcga-tumor | 1316 | 1805 | 1790 | 1805 |
| toxicity | 2807 | 6522 | 6809 | 6772 |
| Average | 1998 | 3829 | 3899 | 3887 |

# F  Ablations on GCondNet

## F.1  Ablation number of steps for decaying the mixing coefficient

Table F.6: We evaluate the impact of decaying the mixing coefficient $\alpha$ for varying number of steps $n_\alpha$. We present the mean±std balanced *validation accuracy* averaged over 25 runs. We find that GCondNet is robust to the decay length $n_\alpha$, and we choose $n_\alpha = 200$ steps for all experiments of this paper unless otherwise specified.

| Dataset | Steps $n_\alpha$ of linear decay for $\alpha$ | | |
|---|---|---|---|
| | 100 | 200 | 400 |
| cll | $88.88 \pm 8.47$ | $88.09 \pm 9.34$ | $88.33 \pm 8.28$ |
| lung | $96.77 \pm 5.07$ | $97.27 \pm 4.97$ | $97.36 \pm 4.76$ |
| meta-p50 | $98.56 \pm 3.70$ | $98.56 \pm 3.70$ | $97.74 \pm 5.02$ |
| meta-dr | $71.20 \pm 8.80$ | $68.36 \pm 10.63$ | $71.92 \pm 8.39$ |
| prostate | $95.56 \pm 7.35$ | $93.97 \pm 10.22$ | $95.11 \pm 7.99$ |
| smk | $76.39 \pm 11.59$ | $76.89 \pm 13.10$ | $75.78 \pm 11.64$ |
| tcga-survival | $69.20 \pm 11.38$ | $70.6 \pm 9.35$ | $68.53 \pm 10.01$ |
| tcga-tumor | $62.06 \pm 13.64$ | $62.46 \pm 18.05$ | $66.13 \pm 15.17$ |
| toxicity | $98.75 \pm 3.12$ | $98.25 \pm 3.83$ | $98.5 \pm 3.73$ |

Table F.7: Statistical analysis of the number of iterations from Table F.6. We report the p-values of a Wilcoxon signed-rank test (Demšar, 2006; Benavoli et al., 2016). The results support our observation, namely that GCondNet is robust to the hyper-parameter $n_\alpha$.

| $n_\alpha = 100$ vs. $n_\alpha = 200$ | $n_\alpha = 100$ vs. $n_\alpha = 400$ | $n_\alpha = 200$ vs. $n_\alpha = 400$ |
|---|---|---|
| 4.00E-01 | 9.44E-01 | 4.96E-01 |

## F.2  Vary the size of backbone's first layer

The GNN component of GCondNet outputs a matrix $\boldsymbol{W}_{\mathrm{GNN}} \in \mathbb{R}^{K \times D}$, where $K$ is the size of the first hidden layer of the underlying backbone network (e.g., an MLP), and $D$ is the number of features. The graph embeddings $\boldsymbol{w}^{(j)} \in \mathbb{R}^K$ described in Algorithm 1 (line 3) match the size $K$ of the backbone MLP model; hence, $K$ is determined by the backbone network, not as a hyper-parameter of GCondNet.

We assess the effect of GCondNet when the backbone network varies in width. We vary the size $K$ of the backbone MLP's first layer and apply GCondNet (with KNN graphs), which computed graph embeddings of size $K$ accordingly. Figure F.1 and Table F.8 demonstrate that GCondNet consistently enhances performance across different values of $K$. Notably, GCondNet greatly improves stability, yielding comparable outcomes across various network widths, unlike the MLP, which exhibits substantial performance decrease in narrower networks.

Table F.8: Numerical results for Figure F.1, comparing the performance when varying the size of the first hidden layer of an MLP, and GCondNet with an identical MLP backbone.

| Dataset | gli | | allaml | | cll | | glioma | |
|---|---|---|---|---|---|---|---|---|
| Size first layer | MLP | GCondNet | MLP | GCondNet | MLP | GCondNet | MLP | GCondNet |
| 20 | $75.38_{\pm13.97}$ | $84.43_{\pm10.84}$ | $78.18_{\pm18.09}$ | $96.80_{\pm5.57}$ | $67.97_{\pm7.43}$ | $81.21_{\pm7.53}$ | $66.00_{\pm17.46}$ | $78.00_{\pm14.26}$ |
| 50 | $79.98_{\pm10.44}$ | $84.02_{\pm9.58}$ | $95.07_{\pm7.39}$ | $97.58_{\pm4.13}$ | $73.49_{\pm4.66}$ | $81.29_{\pm6.93}$ | $56.50_{\pm16.93}$ | $74.50_{\pm14.25}$ |
| 100 | $77.72_{\pm15.30}$ | $85.02_{\pm9.00}$ | $91.30_{\pm6.70}$ | $96.18_{\pm4.90}$ | $78.30_{\pm9.00}$ | $80.70_{\pm5.50}$ | $73.00_{\pm14.90}$ | $76.67_{\pm12.90}$ |
| 150 | $79.88_{\pm15.32}$ | $84.92_{\pm10.12}$ | $91.64_{\pm10.22}$ | $95.16_{\pm6.73}$ | $81.06_{\pm7.82}$ | $80.60_{\pm6.32}$ | $71.50_{\pm15.67}$ | $67.33_{\pm17.12}$ |
| 200 | $81.03_{\pm12.17}$ | $85.66_{\pm10.04}$ | $91.89_{\pm10.50}$ | $96.58_{\pm4.73}$ | $80.11_{\pm6.94}$ | $81.28_{\pm6.37}$ | $74.33_{\pm11.83}$ | $80.83_{\pm11.60}$ |

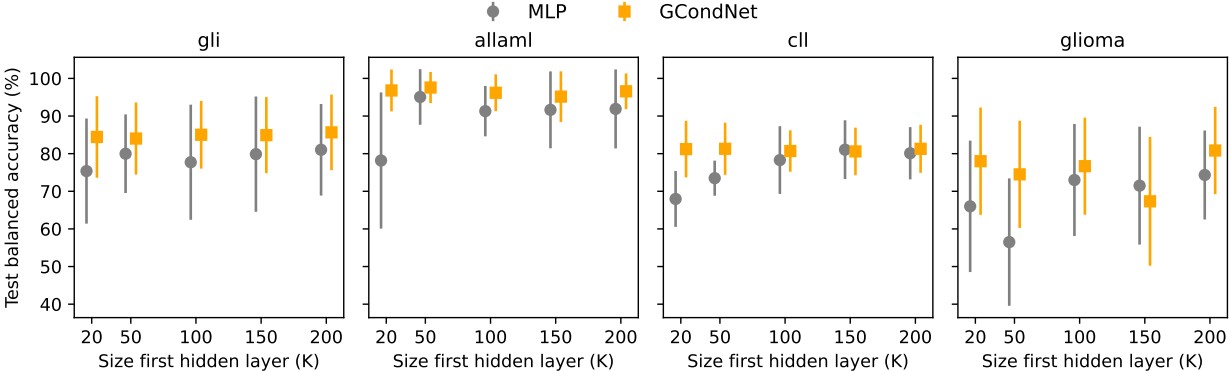

Figure F.1: Comparing the performance when varying the size of the first hidden layer of an MLP, and GCondNet with an identical MLP backbone. GCondNet consistently enhances performance across different values of $K$, and it greatly improves stability, yielding comparable outcomes across various network widths.

### F.3 Ablation on GCondNet's graph construction method and the MLP weight initialisation

**RandEdge Graphs** from Section 3.2. The proposed SRD method creates graphs that contain, on average, 8% of the edges of a fully connected graph (as shown in Appendix B). To create RandEdge graphs with similar statistics, we sample a proportion $p$ from all possible edges in the graph, where $p \sim \mathcal{N}(\mu = 0.08, \sigma = 0.03)$ is sampled for each of the $D$ graphs. We used the same initial node embeddings from Section 2.1. We sample each graph five times and train each of the 25 models on all graphs – resulting in 125 trained models on RandEdge graphs.

**Specialised initialisations** from Section 3.2. We first compute $\boldsymbol{W}_{\text{MLP}}^{[1]}$ using any of the three initialisation methods that we propose below. To mitigate the risk of exploding gradients, we adopt the method proposed by He et al. (He et al., 2015) to rescale the weights. After computing $\boldsymbol{W}_{\text{MLP}}^{[1]}$, we then perform zero-centring on each row of $\boldsymbol{W}_{\text{MLP}}^{[1]}$ and subsequently rescale it to match the standard deviation of the Kaiming initialisation (He et al., 2015). The resulting matrix is used to initialise the first layer of the predictor MLP.

1. **Principal Component Analysis (PCA) initialisation.** We use PCA to compute feature embeddings $\boldsymbol{e}_{\text{PCA}}^{(j)}$ for all features $j$. These embeddings are then concatenated horizontally to form the weight matrix of the first layer of the MLP predictor $\boldsymbol{W}_{\text{MLP}}^{[1]} = [\boldsymbol{e}_{\text{PCA}}^{(1)}, \boldsymbol{e}_{\text{PCA}}^{(2)}, ..., \boldsymbol{e}_{\text{PCA}}^{(D)}]$.

2. **Non-negative matrix factorisation (NMF) initialisation.** NMF has been applied in bioinformatics to cluster gene expression (Kim & Park, 2007; Taslaman & Nilsson, 2012) and identify common cancer mutations (Alexandrov et al., 2013). It approximates $\boldsymbol{X} \approx \boldsymbol{WH}$, with the intuition that the column space of $\boldsymbol{W}$ represents "eigengenes", and the column $\boldsymbol{H}_{:,j}$ represents coordinates of gene $j$ in the space spanned by the eigengenes. The feature embedding is $\boldsymbol{e}_{\text{NMF}}^{(j)} := \boldsymbol{H}_{:,j}$. These embeddings are then concatenated horizontally to form the weight matrix of the first layer of the MLP predictor $\boldsymbol{W}_{\text{MLP}}^{[1]} = [\boldsymbol{e}_{\text{NMF}}^{(1)}, \boldsymbol{e}_{\text{NMF}}^{(2)}, ..., \boldsymbol{e}_{\text{NMF}}^{(D)}]$.

3. **Weisfeiler-Lehman (WL) initialisation.** The WL algorithm (Weisfeiler & Leman, 1968), often used in graph theory, is a method to check whether two given graphs are isomorphic, i.e., identical up to a renaming of the vertices. The algorithm creates a graph embedding of size $N$. For our use-case, the embeddings must have size $K$, the size of the first hidden layer of the predictor MLP; thus, we obtain the feature embeddings $\boldsymbol{e}_{\text{WL}}^{(j)}$ by computing the histogram with $K$ bins of the WL-computed graph embedding, which is then normalised to be a probability density. We apply the WL algorithm on the SRD graphs described in Section 2.1, and finally obtain $\boldsymbol{W}_{\text{WL}}^{[1]} = [\boldsymbol{e}_{\text{WL}}^{(1)}, \boldsymbol{e}_{\text{WL}}^{(2)}, ..., \boldsymbol{e}_{\text{WL}}^{(D)}]$.

Table F.9: We evaluate the robustness of GCondNet on various graph construction methods and compare it to an identically structured and trained MLP initialised with three weight initialisation methods emulating the inductive biases of GCondNet, but without training a GNN. Here, all versions of MLPs and GCondNet use a fixed dropout $p = 0.2$. We report the mean $\pm$ std of the test balanced accuracy averaged over 25 runs and include statistical tests in Table F.10. Results show that GCondNet is robust to the graphs and consistently outperforms a standard MLP across all graph construction methods. Further, GCondNet often outperforms other initialisation methods, highlighting the effectiveness of the GNN-extracted latent structure.

| | MLP | MLP with specialised initialisations | | | GCondNet with different graphs | | |
|---|---|---|---|---|---|---|---|
| | | PCA | NMF | WL | RandEdge | KNN | SRD |
| allaml | $91.30_{\pm6.74}$ | $95.49_{\pm5.97}$ | $95.84_{\pm5.42}$ | $95.33_{\pm5.81}$ | $96.39_{\pm4.89}$ | $96.18_{\pm4.85}$ | $96.36_{\pm4.70}$ |
| cll | $78.30_{\pm8.99}$ | $79.92_{\pm6.48}$ | $78.59_{\pm6.64}$ | $79.98_{\pm6.59}$ | $81.36_{\pm5.78}$ | $80.70_{\pm5.47}$ | $81.54_{\pm7.15}$ |
| gli | $77.72_{\pm15.33}$ | $83.82_{\pm12.35}$ | $85.58_{\pm9.58}$ | $83.79_{\pm10.77}$ | $85.94_{\pm8.65}$ | $85.51_{\pm8.96}$ | $86.36_{\pm8.05}$ |
| glioma | $73.00_{\pm14.88}$ | $75.00_{\pm15.17}$ | $74.67_{\pm13.55}$ | $75.50_{\pm13.41}$ | $75.67_{\pm12.09}$ | $76.67_{\pm12.90}$ | $77.50_{\pm8.51}$ |
| lung | $94.20_{\pm4.95}$ | $96.04_{\pm4.00}$ | $95.05_{\pm4.06}$ | $94.56_{\pm5.97}$ | $94.86_{\pm4.58}$ | $94.68_{\pm4.25}$ | $95.34_{\pm4.49}$ |
| meta-dr | $59.56_{\pm5.50}$ | $55.75_{\pm8.27}$ | $59.36_{\pm6.84}$ | $58.69_{\pm7.36}$ | $57.89_{\pm8.76}$ | $59.34_{\pm8.93}$ | $58.24_{\pm6.36}$ |
| meta-p50 | $94.31_{\pm5.39}$ | $94.70_{\pm4.93}$ | $95.09_{\pm4.80}$ | $95.81_{\pm5.05}$ | $95.86_{\pm4.25}$ | $96.37_{\pm4.00}$ | $96.26_{\pm3.79}$ |
| prostate | $88.76_{\pm5.55}$ | $91.04_{\pm5.08}$ | $89.36_{\pm6.49}$ | $89.97_{\pm5.94}$ | $89.56_{\pm6.37}$ | $90.38_{\pm5.59}$ | $89.96_{\pm6.14}$ |
| smk | $64.42_{\pm8.44}$ | $66.79_{\pm10.80}$ | $65.87_{\pm7.35}$ | $64.47_{\pm8.17}$ | $66.13_{\pm8.12}$ | $65.92_{\pm8.68}$ | $68.08_{\pm7.31}$ |
| tcga-survival | $56.28_{\pm6.73}$ | $55.22_{\pm7.39}$ | $60.08_{\pm6.18}$ | $54.79_{\pm8.23}$ | $58.31_{\pm7.81}$ | $58.61_{\pm7.01}$ | $56.36_{\pm9.41}$ |
| tcga-tumor | $48.19_{\pm7.75}$ | $50.95_{\pm10.59}$ | $51.49_{\pm9.77}$ | $49.67_{\pm8.86}$ | $51.57_{\pm9.10}$ | $51.70_{\pm8.82}$ | $52.43_{\pm7.57}$ |
| toxicity | $93.21_{\pm6.14}$ | $92.58_{\pm5.46}$ | $89.11_{\pm5.99}$ | $92.49_{\pm5.70}$ | $95.06_{\pm4.17}$ | $95.22_{\pm3.93}$ | $95.25_{\pm4.54}$ |
| Average rank | 6.08 | 4.42 | 4.33 | 5.17 | 3.17 | 2.75 | **2.08** |

Table F.10: Statistical analysis of Table F.9. We compare both GCondNet w/ SRD and KNN variants to each other, to GCondNet with RandomEdge graphs, and to the three weight initialisations methods we proposed. We perform statistical testing using the Wilcoxon test (Demšar, 2006; Benavoli et al., 2016). The results support our discussion that GCondNet is robust to the graph construction method, as the performance differences between different graph constructing methods are not significant at $\alpha = 0.05$. Furthermore, we find that both GCondNet variants (SRD and KNN) perform significantly better than the MLP baseline and the PCA and WL initialisations.

| Model A | vs. | Model B | Wilcoxon p-value |
|---|---|---|---|
| GCondNet (KNN) | | MLP | 9.7656e-04 |
| GCondNet (KNN) | | MLP (NMF) | 1.0934e-01 |
| GCondNet (KNN) | | MLP (PCA) | 3.418e-02 |
| GCondNet (KNN) | | MLP (WL) | 4.8828e-04 |
| GCondNet (SRD) | | MLP | 3.418e-03 |
| GCondNet (SRD) | | MLP (NMF) | 1.2939e-01 |
| GCondNet (SRD) | | MLP (PCA) | 1.6113e-02 |
| CondNet (SRD) | | MLP (WL) | 3.418e-03 |
| GCondNet (SRD) | | GCondNet (KNN) | 6.8894e-01 |
| GCondNet (SRD) | | GCondNet (RandEdge) | 1.2939e-01 |
| GCondNet (KNN) | | GCondNet (RandEdge) | 5.9335e-01 |

# G GCondNet beyond small-sample and high-dimensional data

### G.1 Varying the dataset size N

We assess the scalability of GCondNet in comparison to its backbone model. For this evaluation, we use an MLP backbone and train a GCondNet with KNN graphs. Both GCondNet and the baseline MLP are trained under the same conditions using the same dataset splits.

**Setup:** Here, we adopt a different evaluation setup from the main paper to facilitate a fair and robust comparison across various dataset splits. Specifically, we examine two datasets from Metabric, which contains 2,000 samples, and two from TCGA, which contains 500 samples. We reserve 20% of the total data for testing, creating a test set of 400 samples for Metabric and 100 samples for TCGA. We use this fixed test set to evaluate the models across training subsets of varying sizes. We use the remaining 80% of the data to create training sets of different sizes. Specifically, we reduce the training set size by sequentially decreasing the number of samples from the larger preceding subsets without introducing new samples. We conduct five-fold cross-validation for each resulting training subset, training on four folds and using the other fold for validation.

**Results:** In terms of accuracy, Figure G.2 (numerical results in Table G.11) illustrates the test accuracy when increasing the training dataset size $N$. GCondNet consistently outperforms the MLP on datasets prone to high overfitting, such as 'tcga-survival', 'tcga-tumor', and 'meta-dr', with average performance improvements of 5.7%. On 'meta-pam50', the MLP occasionally outperforms GCondNet; however, the predictive performance on this dataset is already high and nearly saturated. Overall, GCondNet proves effective and can enhance performance beyond very small datasets.

In terms of training time, Figure G.3 (numerical results in Table G.11) indicates that GCondNet sometimes exhibits a slightly longer convergence time, primarily due to the complexities involved in training the auxiliary GNN component and generally slower convergence. However, the increase in runtime does not disproportionately escalate with larger datasets. The overhead remains relatively constant across varying dataset sizes, averaging just 20% more training time, which indicates that GCondNet scales effectively without significant size-dependent overhead, while providing performance improvements.

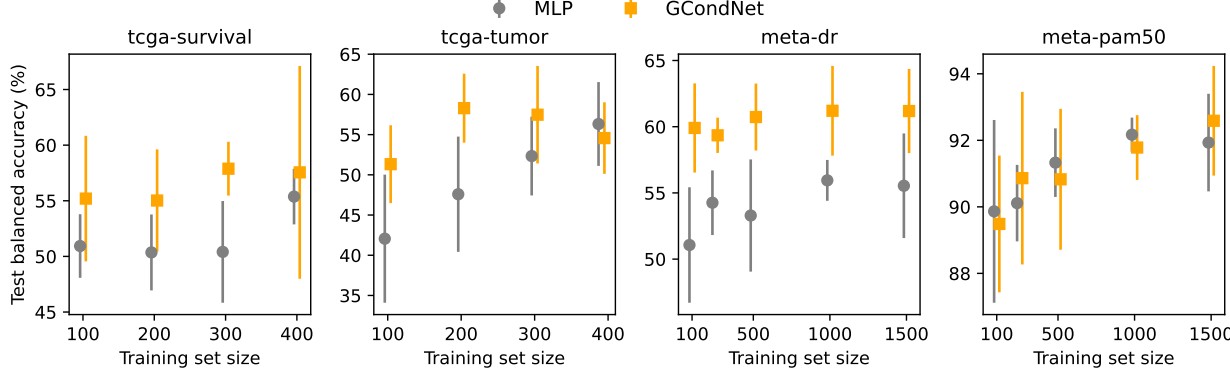

Figure G.2: Test accuracy when increasing the training dataset size $N$, showing the mean±std across four datasets. GCondNet outperforms the MLP on datasets prone to high overfitting, such as 'tcga-survival', 'tcga-tumor', and 'meta-dr'. On 'meta-pam50', the MLP sometimes outperforms GCondNet; however, note that the predictive performance on this dataset is already high and nearly saturated. Overall, GCondNet proves effective and can enhance performance beyond very small datasets.

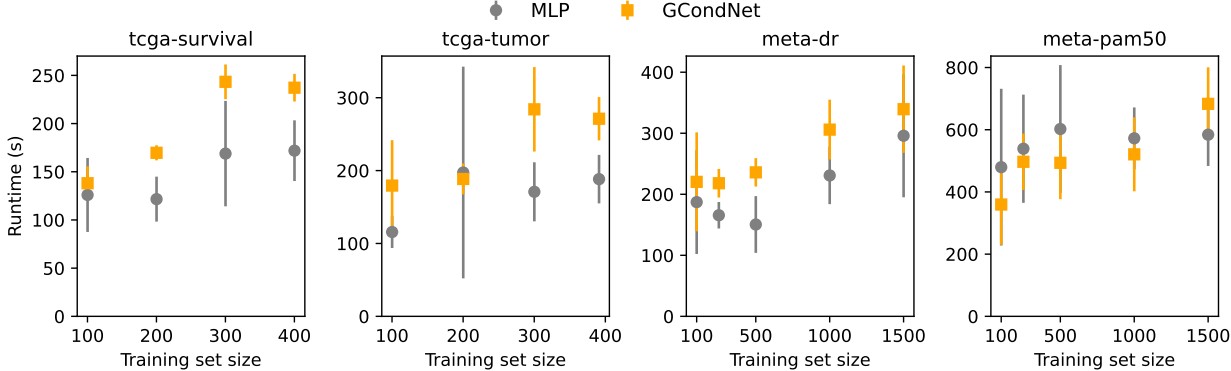

Figure G.3: Training time with increasing dataset size, showing the mean±std across four datasets. GCondNet generally requires slightly more time to converge, but the overhead is relatively constant across different dataset sizes, indicating that GCondNet scales well and does not incur significant size-dependent overhead.

Table G.11: Comparison of MLP and GCondNet (with an MLP backbone) on datasets of varying sizes ($N$): We report the mean±std of test balanced accuracy (%) and the average training time over five runs. GCondNet generally improves the baseline MLP's performance, though it introduces some computational overhead which is generally constant independent of the dataset size.

| Dataset | Dataset Size (N) | Model | Test Accuracy | Runtime (m) |
|---------|------------------|-------|---------------|-------------|
| tcga-survival | 100 | MLP | $50.93_{\pm2.86}$ | 2.1 |
| | | GCondNet | $55.20_{\pm5.64}$ | 2.3 |
| | 200 | MLP | $50.36_{\pm3.41}$ | 2.0 |
| | | GCondNet | $55.02_{\pm4.59}$ | 2.8 |
| | 300 | MLP | $50.41_{\pm4.57}$ | 2.8 |
| | | GCondNet | $57.88_{\pm2.42}$ | 4.0 |
| | 400 | MLP | $55.38_{\pm2.50}$ | 2.9 |
| | | GCondNet | $57.55_{\pm9.56}$ | 4.0 |
| tcga-tumor | 100 | MLP | $42.06_{\pm7.96}$ | 1.9 |
| | | GCondNet | $51.32_{\pm4.84}$ | 3.0 |
| | 200 | MLP | $47.59_{\pm7.17}$ | 3.3 |
| | | GCondNet | $58.29_{\pm4.29}$ | 3.2 |
| | 300 | MLP | $52.33_{\pm4.90}$ | 2.8 |
| | | GCondNet | $57.48_{\pm6.06}$ | 4.7 |
| | 400 | MLP | $56.32_{\pm5.21}$ | 3.1 |
| | | GCondNet | $54.57_{\pm4.45}$ | 4.5 |
| meta-dr | 100 | MLP | $51.07_{\pm4.36}$ | 3.1 |
| | | GCondNet | $59.91_{\pm3.37}$ | 3.7 |
| | 250 | MLP | $54.26_{\pm2.44}$ | 2.8 |
| | | GCondNet | $59.35_{\pm1.33}$ | 3.6 |
| | 500 | MLP | $53.29_{\pm4.24}$ | 2.5 |
| | | GCondNet | $60.73_{\pm2.53}$ | 3.9 |
| | 1000 | MLP | $55.95_{\pm1.54}$ | 3.8 |
| | | GCondNet | $61.20_{\pm3.39}$ | 5.1 |
| | 1500 | MLP | $55.54_{\pm3.95}$ | 4.9 |
| | | GCondNet | $61.18_{\pm3.18}$ | 5.6 |
| meta-pam | 100 | MLP | $89.86_{\pm2.75}$ | 8.0 |
| | | GCondNet | $89.49_{\pm2.06}$ | 6.0 |
| | 250 | MLP | $90.11_{\pm1.15}$ | 9.0 |
| | | GCondNet | $90.86_{\pm2.59}$ | 8.3 |
| | 500 | MLP | $91.33_{\pm1.03}$ | 10.0 |
| | | GCondNet | $90.83_{\pm2.12}$ | 8.2 |
| | 1000 | MLP | $92.17_{\pm0.51}$ | 9.5 |
| | | GCondNet | $91.78_{\pm0.97}$ | 8.7 |
| | 1500 | MLP | $91.93_{\pm1.47}$ | 9.7 |
| | | GCondNet | $92.59_{\pm1.65}$ | 11.4 |

## G.2 Vary the sample-to-feature ratio N/D

Assessing the impact of the sample-to-feature ratio $N/D$: Table G.12 presents the complete numerical results, while Figure G.4 visually represents these results, highlighting that GCondNet improves both accuracy and stability across various $N/D$ sizes.

To isolate the impacts of GCondNet's inductive bias and regularisation, we compare the performance of GCondNet (with an MLP backbone and KNN graphs) to an identical MLP baseline. Both models were trained under the same conditions, as presented in the paper. We select four datasets ("allaml", "cll", "gli", "glioma") to examine a broad range of $N/D$ ratios (0.01, 0.05, 0.2, 1, 5), keeping the per-dataset number of samples $N$ constant (presented in Appendix C). We adjust the number of features $D$ accordingly–for instance, a dataset with $N = 100$ samples had $D$ reduced from 10,000 to 20. Note that this reduction led to a performance loss for both models, which is expected, as reducing the feature set might also remove important features. However, here we are interested in the relative performance of GCondNet w.r.t to the baseline MLP.

We took several measures to ensure robust results:

1. We vary the $N/D$ ratio by changing the size of $D$. Particularly, increasing the ratio is done by reducing the feature subset $D$, where smaller feature subsets were derived by removing features from the preceding larger subsets (e.g., $[d_1, d_2, d_3, d_4] \rightarrow [d_1, d_2, d_4] \rightarrow [d_1, d_2] \rightarrow [d_2]$), ensuring that no new features were introduced during the reduction process.

2. We created feature subsets for each dataset separately, repeating the process five times.

3. Both GCondNet and the MLP were trained using the same feature sets. For each configuration – encompassing model, dataset, feature-subset, and feature-subset-repeat– we trained 25 distinct models (via 5-fold cross-validation, repeated 5 times), resulting in 125 runs per entry in our results table. Overall, this setup led to the training and evaluation of 5,000 models in total.

Notably, GCondNet demonstrates greater stability and lower performance variability, even with reduced feature sets (Figure G.4). For instance, in the "cll" dataset at $N/D$ ratios of 0.05 and 0.2, GCondNet exhibits smooth performance transitions, contrasting with the fluctuating trends observed in the MLP baseline. These findings underscore the effectiveness of GCondNet's inductive bias across diverse datasets and feature configurations.

Table G.12: Assessing the impact of the sample-to-feature ratio $N/D$. We presented the mean$\pm$ std test balanced accuracy (%) of an MLP and a GCondNet with an equivalent MLP backbone, averaged over 125 runs. We adjust the $N/D$ ratio by holding the number of samples $N$ constant and varying the number of features $D$. The results show that GCondNet's inductive bias predominantly enhances performance across different $N/D$ ratios by up to 10%, except for "glioma" when $N/D \in \{1, 5\}$, where it slightly reduces accuracy.

| Dataset | gli | | allaml | | cll | | glioma | |
|---|---|---|---|---|---|---|---|---|
| N/D ratio | MLP | GCondNet | MLP | GCondNet | MLP | GCondNet | MLP | GCondNet |
| 0.01 | $82.30_{\pm10.28}$ | $84.46_{\pm8.11}$ | $91.67_{\pm8.94}$ | $96.61_{\pm4.62}$ | $78.38_{\pm7.09}$ | $81.03_{\pm5.75}$ | $73.13_{\pm12.77}$ | $76.13_{\pm12.76}$ |
| 0.05 | $80.52_{\pm12.89}$ | $83.91_{\pm10.48}$ | $85.28_{\pm12.08}$ | $94.65_{\pm6.72}$ | $72.23_{\pm9.50}$ | $80.76_{\pm6.23}$ | $72.67_{\pm14.23}$ | $75.60_{\pm10.20}$ |
| 0.20 | $78.57_{\pm12.56}$ | $81.77_{\pm10.59}$ | $84.99_{\pm11.91}$ | $90.05_{\pm8.79}$ | $72.95_{\pm8.48}$ | $78.49_{\pm6.52}$ | $69.40_{\pm17.58}$ | $72.57_{\pm13.42}$ |
| 1.00 | $74.14_{\pm13.43}$ | $75.97_{\pm13.30}$ | $74.85_{\pm13.85}$ | $81.22_{\pm12.52}$ | $68.02_{\pm7.88}$ | $68.02_{\pm8.66}$ | $58.37_{\pm17.85}$ | $57.67_{\pm17.05}$ |
| 5.00 | $59.70_{\pm13.49}$ | $64.98_{\pm13.51}$ | $61.89_{\pm15.34}$ | $66.13_{\pm14.83}$ | $56.59_{\pm14.15}$ | $60.93_{\pm10.95}$ | $52.83_{\pm16.34}$ | $49.97_{\pm16.12}$ |

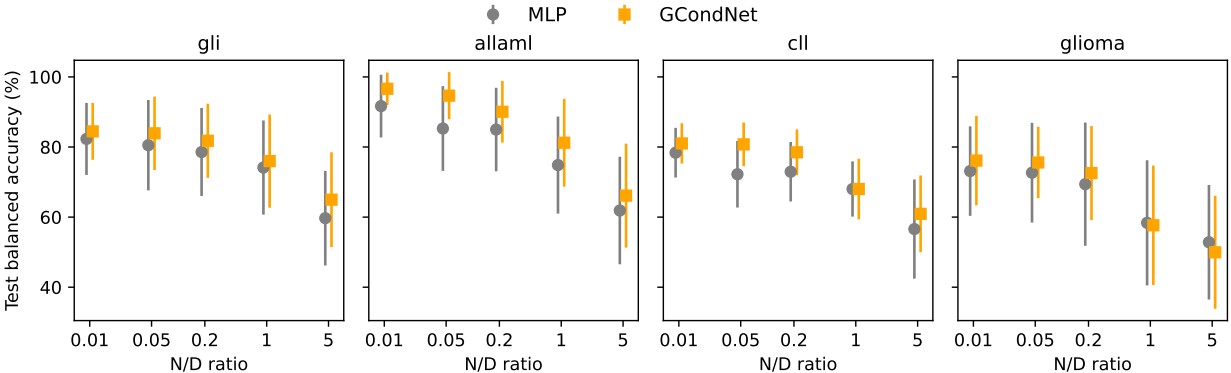

Figure G.4: Assessing the impact of the sample-to-feature ratio $N/D$. We presented the mean$\pm$ std test balanced accuracy (%) of an MLP and a GCondNet with an equivalent MLP backbone, averaged over 125 runs. We adjust the $N/D$ ratio by holding the number of samples $N$ constant and varying the number of features $D$. The results show that GCondNet's inductive bias predominantly enhances performance across different $N/D$ ratios by up to 10%, except for "glioma" when $N/D \in \{1, 5\}$, where it slightly reduces accuracy.

