# OpenReview forum: "GCondNet: A Novel Method for Improving Neural Networks on Small High-Dimensional Tabular Data"
_TMLR — Accepted by TMLR_

### Review · Reviewer_x2re · 2024-06-06

**Summary Of Contributions:**

The paper presents GCondNet an approach for using implicit structure in the tabular data for improving neural network training. This proposed approach is designed for scenarios where small number tabular data is available with large number of features. The key idea is to introduce the inductive bias in the model’s training by ensuring similar features have similar coefficients early in the training. Many small graphs are built as a GNN initially and the similarity information from the features flows through the GNN to the model being trained. The evaluation shows that this approach  is quite effective on several benchmarks compared to large number of baselines.

**Audience:**

Yes

**Claims And Evidence:**

Yes

**Requested Changes:**

Please see questions above.

**Strengths And Weaknesses:**

Strengths:
1. The paper is well-written and the presented ideas are easy to follow
2. The idea presented in the paper is quite novel and shown to effective empirically.
3. The evaluation is extensive and shows improvements over several baselines. The study on inductive bias compared to other initialization methods such as NMF and PCA is quite interesting. The proposed approach is shown to be effective when used in combination with multiple architectures KNN, SRD, Transformers, etc.

Questions:
1. I’m curious to see at what value of N/D the method will become less effective. A study on increasing the value N/D until GCondNet is not useful will be quite interesting.
2. Training time for GCondNet is longer than other baselines considered. Would training a existing baselines  longer or baselines  with larger number of parameters be better in practice if given equal training time?

---

> ### Author Response · Authors · 2024-06-21
> **Response by Authors**
>
> # Response-`x2re` (part 1/2)
>
> Thank you for your positive feedback and constructive suggestions! We have addressed all your questions below, and we have updated the manuscript with this discussion and the new results.
>
> ## > Q1: The effect of training GCondNet while varying the ratio N/D
>
> We conducted further experiments to examine the effect of the sample-to-feature ratio $N/D$ on GCondNet's performance. The results presented below, based on training over 5,000 models, show that GCondNet consistently improves the performance and stability across tasks with diverse sample-to-feature ratios $N/D$.
>
> **Setup:** We compare the performance of GCondNet (with an MLP backbone and KNN graphs) to an identical MLP baseline. Both models were trained under the same conditions, as presented in the paper. We selected four datasets (allaml, cll, gli, glioma) to examine a broad range of $N/D$ ratios (0.01, 0.05, 0.2, 1, 5), keeping the per-dataset number of samples $N$ constant. We adjusted the number of features $D$ accordingly—for instance, a dataset with $N = 100$ samples had $D$ reduced from 10,000 to 20. Note that this reduction led to a performance loss for both models, which is expected, as reducing the feature set might also remove important features. However, here, we are interested in the relative performance of GCondNet w.r.t to the baseline MLP.
>
> We took several measures to ensure robust results:
>
> 1. We vary the $N/D$ ratio by changing the size of $D$. In particular, we increase the ratio by reducing the feature subset $D$, where smaller feature subsets were derived by removing features from the preceding larger subsets (e.g., $[d_1, d_2, d_3, d_4] \rightarrow [d_1, d_2, d_4] \rightarrow [d_1, d_2] \rightarrow [d_2]$): this ensures that no new features were introduced during the reduction process.
>
> 2. We created feature-subsets for each dataset separately, repeating the process five times.
>
> 3. Both GCondNet and the MLP were trained using the same feature sets. For each configuration – encompassing model, dataset, feature-subset, and feature-subset-repeat– we trained 25 distinct models (via 5-fold cross-validation, repeated 5 times), resulting in 125 runs per entry in our results table. Overall, this setup led to the training and evaluation of 5,000 models in total.
>
> **Results**: Table `R1` shows GCondNet predominantly enhances performance across different $N/D$ ratios by up to 10%, except for “glioma” with $N/D$ equal to 1 or 5, where it slightly reduces accuracy. Additionally, GCondNet shows greater stability, evidenced by lower standard deviations and consistent performance trends, even with fewer features. For instance, on the “cll” dataset, GCondNet demonstrates a smooth performance transition, unlike MPL, which exhibits a reverse trend in performance at ratios of 0.05 and 0.2. In conclusion, the results demonstrate that GCondNet is effective and scales across varying $N/D$ ratios.
>
> `Changes:` We have updated the manuscript with the following changes: (i) we added a summary of this discussion and results in Section 3.2, and (ii) we created Appendix G.2, which includes the complete numerical results for each dataset, together with illustrative plots.
>
> *Table `R1` (same as Table G.12 in the updated manuscript):* The mean ± std (averaged over 125 runs) of the test balanced accuracy of GCondNet and an equivalent MLP when varying the sample-to-feature ratio $N/D$.
>
> | Dataset | gli | gli | allaml | allaml | cll | cll | glioma | glioma |
> | --- | --- | --- | --- | --- | --- | --- | --- | --- |
> | **N/D ratio** | MLP | GCondNet | MLP | GCondNet | MLP | GCondNet | MLP | GCondNet |
> | 0.01 | 82.30 ± 10.28 | 84.46 ± 8.11 | 91.67 ± 8.94 | 96.61 ± 4.62 | 78.38 ± 7.09 | 81.03 ± 5.75 | 73.13 ± 12.77 | 76.13 ± 12.76 |
> | 0.05 | 80.52 ± 12.89 | 83.91 ± 10.48 | 85.28 ± 12.08 | 94.65 ± 6.72 | 72.23 ± 9.50 | 80.76 ± 6.23 | 72.67 ± 14.23 | 75.60 ± 10.20 |
> | 0.20 | 78.57 ± 12.56 | 81.77 ± 10.59 | 84.99 ± 11.91 | 90.05 ± 8.79 | 72.95 ± 8.48 | 78.49 ± 6.52 | 69.40 ± 17.58 | 72.57 ± 13.42 |
> | 1.00 | 74.14 ± 13.43 | 75.97 ± 13.30 | 74.85 ± 13.85 | 81.22 ± 12.52 | 68.02 ± 7.88 | 68.02 ± 8.66 | 58.37 ± 17.85 | 57.67 ± 17.05 |
> | 5.00 | 59.70 ± 13.49 | 64.98 ± 13.51 | 61.89 ± 15.34 | 66.13 ± 14.83 | 56.59 ± 14.15 | 60.93 ± 10.95 | 52.83 ± 16.34 | 49.97 ± 16.12 |

---

> ### Author Response · Authors · 2024-06-21
> **Response by Authors**
>
> # Response-`x2re` (part 2/2)
>
> ## > Q2: Clarification on the time required for tuning GCondNet
>
> GCondNet is leveraging a backbone model, such as an MLP, while retaining the same training hyper-parameters as the optimised backbone model. Consequently, the time needed to tune GCondNet is typically equivalent to tuning the backbone model.
>
> In the paper, we mainly use an MLP as the backbone of GCondNet, but other backbones (such as a Transformer) are possible. We initially performed hyperparameter tuning on an MLP. After selecting the optimal hyperparameters, we retrained GCondNet with the same MLP backbone, using two graph construction methods, resulting in two variants: GCondNet (KNN) and GCondNet (SRD), as shown in Table 1. It is important to note that GCondNet did not undergo additional separate hyperparameter tuning; its integration required only: (i) creating the graphs once per dataset, which adds a negligible computational overhead; (ii) optionally tuning the GNN in GCondNet (although the initial settings performed well and we did not pursue specialised tuning); and (iii) retraining the backbone MLP model, which requires marginal additional time beyond the initial tuning of the MLP. Our findings in Figure 2 and Figure 3 demonstrate that GCondNet enhances performance compared to an already well-tuned backbone model with minimal computational overhead, leading to the improved results noted in Table 1.
>
> `Changes:` We have made updates to the manuscript: (i) In Appendix D, we clarified that we intended GCondNet to be used while retaining the same training hyperparameters as the optimised backbone model. (ii) In Appendix E, we stated that the computation cost added by GCondNet is a one-time training cost incurred after the backbone has been tuned.
>
> Thank you again for your thoughtful review! We believe that these clarifications have strengthened the manuscript.

---

### Review · Reviewer_nQny · 2024-06-13

**Summary Of Contributions:**

This paper introduces a framework called GCondNet for enhancing neural networks by leveraging implicit relationships in high-dimensional tabular data. It constructs multiple graphs corresponding to each data dimension and utilizes a Graph Neural Network to condition the initial layer parameters of the base predictor model. This strategy improves performance and robustness across various real-world datasets, particularly in scenarios with limited samples but numerous features. The approach is generic, not requiring external knowledge graphs, which broadens its applicability across different domains.

**Audience:**

Yes

**Claims And Evidence:**

Yes

**Requested Changes:**

1. The claim at the end of the first paragraph in Section 2.1, stating that graph construction for each task takes five seconds, appears to be based on a specific experimental setup not yet introduced. Please clarify whether this timing is general or pertains to a particular dataset described later, and if so, detail this setup earlier to avoid confusion.

2. The paper notes increased training times for GCondNet compared to benchmark methods. Please explain the reasons for this slower performance. Additionally, it would be helpful to discuss how the method scales with larger datasets—does the increase in the number of samples or features exacerbate the training time disproportionately?

3. Given the method’s outlined suitability for datasets with many features but fewer samples, could you provide examples or scenarios where GCondNet underperforms? Specifically, insights into its performance on datasets with a large number of samples or fewer features would be valuable for understanding its limitations.

4. The determination of the parameter $K$ in the graph neural network setup remains unclear. Could you elaborate on how $K$ is chosen and discuss whether the model’s performance is sensitive to variations in $K$? This information would aid in assessing the robustness and applicability of GCondNet across different settings.

**Strengths And Weaknesses:**

[Strengths]

1. The novel use of implicit sample-wise relationships and graph-based regularization provides a clear improvement over existing methods.

2. The method’s effectiveness is well-demonstrated with extensive experiments on multiple real-world datasets, showing significant gains in performance.

3. The paper is clearly written, with a thorough explanation of the methodological innovations and the underlying theoretical considerations.



[Weaknesses]

1. The training time for GCondNet is longer than for benchmark methods, which could limit its practicality in some scenarios.

2. The definitions of “small” and “large” in terms of sample size and feature number are vague and need clarification to better assess the method’s applicability.

---

> ### Author Response · Authors · 2024-06-21
> **Response by Authors**
>
> # Response-`nQny` (part 1/4)
>
> Thank you for the detailed review and constructive feedback! We have addressed all your questions below, and we have updated the manuscript with this discussion and the new results.
>
> ## > RC1: Clarify the timing for the graph construction "taking five seconds to generate all graphs"
>
> The graph construction phase is typically fast, performed once per dataset, and adds a negligible computational overhead. We have removed the "five seconds”, which we initially intended to emphasise that the time required for graph construction is insignificant.
>
> Specifically, we introduced two approaches for graph construction: KNN graphs and SRD graphs.  In terms of constructing KNN graphs for each of the $D$ datasets, each node is connected to the closest $K$ neighbours. The computational complexity for constructing all KNN graphs is $O(D \cdot N \log N + D \cdot N \cdot K)$, where the first term accounts for sorting the features across all graphs and the second term is for creating the edges. Even in our largest datasets, the total number of operations—approximately 20 million (20,000 * 200 * 5)—is negligible for standard CPUs. The SRD graphs typically have a similar number of edges as the KNN graphs (as presented in Appendix B), thus requiring similar time. In conclusion, both our proposed graph construction methods are fast, and have negligible computational cost.
>
> `Changes:` We have updated Section 2.1 by (i) eliminating the specific term "five seconds" to prevent ambiguity and (ii) incorporating the aforementioned discussion to clarify the particular time complexity involved in creating each of the two graph types.

---

> ### Author Response · Authors · 2024-06-21
> **Response by Authors**
>
> # Response-`nQny` (part 2a/4)
>
> > Note: This response is divided into two parts (`2a` and `2b`) because of the character limit.
>
> ## > RC2 / W1: Discuss how GCondNet scales with larger datasets. Does the increase in the number of samples or features exacerbate the training time disproportionately?
>
> GCondNet's training can be slower because, on top of the backbone predictor model, it trains an auxiliary GNN which takes as input the graphs (containing the entire dataset, not just a mini-batch). As we detailed in our response to Q2 for Reviewer x2re, we designed GCondNet to use the same training hyperparameters as the optimised backbone model (e.g., an MLP). GCondNet undergoes a one-off training using the optimal training settings for the backbone model, for which we next discuss the scalability.
>
> We conducted new experiments to assess GCondNet's scalability to larger datasets in comparison to training the backbone model alone. For this evaluation, we use an MLP backbone and train a GCondNet (with KNN graphs). Both GCondNet and the baseline MLP are trained under the same conditions using the same dataset splits. Whilst GCondNet was designed for small datasets, the new results show that GCondNet can also scale effectively to larger datasets without substantially increasing the training time while still maintaining stable training behaviour and providing performance improvements.
>
> **Setup:** For this ablation, we adopt a different evaluation setup from the main paper to facilitate a fair and robust comparison across various dataset splits. Specifically, we focus on two datasets from Metabric, which contains 2,000 samples, and two from TCGA, which contains 500 samples. We reserve 20% of the total data for testing, creating a test set of 400 samples for Metabric and 100 samples for TCGA. We use this fixed test set to evaluate the models across training subsets of varying sizes. We use the remaining 80% of the data to create training sets of different sizes. Specifically, we reduce the training set size by sequentially decreasing the number of samples from the larger preceding subsets without introducing new samples. We conduct five-fold cross-validation for each resulting training subset, training on four folds and using the remaining fold for validation.
>
> The **results** are discussed in part `2b`.

---

> ### Author Response · Authors · 2024-06-21
> **Response by Authors**
>
> # Response-`nQny` (part 2b/4)
>
> Note: This response is divided into two parts (`2a` and `2b`) because of the character limit.
>
>
>
> **Results:** Table `R2` presents the test accuracy when increasing the training dataset size $N$. GCondNet consistently outperforms the MLP on "tcga-survival", "tcga-tumor", and "meta-dr", with average performance improvements of 5.7%. On "meta-pam50", the MLP occasionally outperforms GCondNet; however, the predictive performance on this dataset is already high and nearly saturated. Overall, GCondNet proves effective and can enhance performance beyond very small datasets
>
> In terms of training time, Table `R2` indicates that GCondNet sometimes exhibits a longer convergence time, primarily due to training the auxiliary GNN component, which computes graph embeddings over large graphs. However, the increase in runtime does not disproportionately escalate with larger datasets. The overhead remains relatively constant across varying dataset sizes, averaging 20% longer training time, which indicates that GCondNet is capable of scaling even to larger datasets without significant size-dependent overhead.
>
> In terms of scalability with respect to the number of features, the results in the paper already report on experiments with high-dimensional datasets (with up to 22,283 features). We conducted additional experiments to evaluate GCondNet's performance on smaller feature subsets. The results, derived from training over 5,000 models, demonstrate that GCondNet consistently enhances performance and stability across tasks with various numbers of features. For a more detailed discussion on this aspect, please refer to our response to Q1 for Reviewer x2re.
>
> `Changes:` We have added this discussion in Appendix G.1, together with the new results on GCondNet's scalability on larger datasets and illustrative plots.
>
> *Table `R2` (same as Table G.11 in the updated manuscript): Comparison of MLP and GCondNet (with an MLP backbone) on datasets of varying sizes.*
>
> | Dataset | Dataset Size (N) | Model | Test Accuracy | Runtime (m) |
> | --- | --- | --- | --- | --- |
> | tcga-survival | 100 | MLP | 50.93 ± 2.86 | 2.1 |
> |  |  | GCondNet | 55.20 ± 5.64 | 2.3 |
> |  | 200 | MLP | 50.36 ± 3.41 | 2.0 |
> |  |  | GCondNet | 55.02 ± 4.59 | 2.8 |
> |  | 300 | MLP | 50.41 ± 4.57 | 2.8 |
> |  |  | GCondNet | 57.88 ± 2.42 | 4.0 |
> |  | 400 | MLP | 55.38 ± 2.50 | 2.9 |
> |  |  | GCondNet | 57.55 ± 9.56 | 4.0 |
> | tcga-tumor | 100 | MLP | 42.06 ± 7.96 | 1.9 |
> |  |  | GCondNet | 51.32 ± 4.84 | 3.0 |
> |  | 200 | MLP | 47.59 ± 7.17 | 3.3 |
> |  |  | GCondNet | 58.29 ± 4.29 | 3.2 |
> |  | 300 | MLP | 52.33 ± 4.90 | 2.8 |
> |  |  | GCondNet | 57.48 ± 6.06 | 4.7 |
> |  | 400 | MLP | 56.32 ± 5.21 | 3.1 |
> |  |  | GCondNet | 54.57 ± 4.45 | 4.5 |
> | meta-dr | 100 | MLP | 51.07 ± 4.36 | 3.1 |
> |  |  | GCondNet | 59.91 ± 3.37 | 3.7 |
> |  | 250 | MLP | 54.26 ± 2.44 | 2.8 |
> |  |  | GCondNet | 59.35 ± 1.33 | 3.6 |
> |  | 500 | MLP | 53.29 ± 4.24 | 2.5 |
> |  |  | GCondNet | 60.73 ± 2.53 | 3.9 |
> |  | 1000 | MLP | 55.95 ± 1.54 | 3.8 |
> |  |  | GCondNet | 61.20 ± 3.39 | 5.1 |
> |  | 1500 | MLP | 55.54 ± 3.95 | 4.9 |
> |  |  | GCondNet | 61.18 ± 3.18 | 5.6 |
> | meta-pam | 100 | MLP | 89.86 ± 2.75 | 8.0 |
> |  |  | GCondNet | 89.49 ± 2.06 | 6.0 |
> |  | 250 | MLP | 90.11 ± 1.15 | 9.0 |
> |  |  | GCondNet | 90.86 ± 2.59 | 8.3 |
> |  | 500 | MLP | 91.33 ± 1.03 | 10.0 |
> |  |  | GCondNet | 90.83 ± 2.12 | 8.2 |
> |  | 1000 | MLP | 92.17 ± 0.51 | 9.5 |
> |  |  | GCondNet | 91.78 ± 0.97 | 8.7 |
> |  | 1500 | MLP | 91.93 ± 1.47 | 9.7 |
> |  |  | GCondNet | 92.59 ± 1.65 | 11.4 |

---

> ### Author Response · Authors · 2024-06-21
> **Response by Authors**
>
> # Response-`nQny` (part 3/4)
>
> ## > RC3: Discuss scenarios when GCondNet underperforms
>
> GCondNet is leveraging a backbone model, such as an MLP (or a Transformer), and as such, it inherits some of its limitations. Our results (in Table 1 of the manuscript) show that GCondNet, in general, outperforms many of the benchmarks on many datasets. However, it is likely that it is less performant in scenarios which require additional feature selection mechanisms.
>
> For instance, in the case of the “tcga-survival” dataset, GCondNet (with an MLP backbone) enhances MLP performance by 2%, but it still lags behind top methods like Random Forest, WPFS, and CAE, which exhibit an additional 1-7% better performance. This suggests that the additional feature selection capabilities of these approaches can lead to further improvements. While in this work we did not analyse this behaviour in detail, GCondNet, in principle, can readily handle this. At this point, we envision two scenarios: first, one can implement an additional regularisation in the backbone model or use an architecture that is capable of this. Second, one can also control this in the GNN stage by omitting potentially less informative graphs in the learning stage.  Analysing and addressing these aspects of GCondNet could be a direction for future work, but is beyond the scope of this paper.
>
> `Changes:` We have revised Section 3.1 to incorporate this discussion and to highlight an example where GCondNet does not perform as well.
>
> ## > RC4: Clarify how K is chosen in the graph neural network, and whether GCondNet is sensitive to varying K
>
> The GNN component of GCondNet outputs a matrix $\boldsymbol{W}_{\text{GNN}} \in \mathbb{R}^{K \times D}$, where $K$ is the size of the first hidden layer of the underlying backbone network (e.g., an MLP), and $D$ is the number of features. The graph embeddings $\boldsymbol{w}^{(j)} \in \mathbb{R}^K$ described in Algorithm 1 (line 3) match the size $K$ of the backbone MLP model; hence, $K$ is determined by the backbone network, not as a hyper-parameter of GCondNet.
>
> We conducted new experiments to assess GCondNet's effect when the backbone network varies in width $K$. We vary the size $K$ of the backbone MLP's first layer and apply GCondNet (with KNN graphs), which computed graph embeddings of size $K$ accordingly. The new results Table `R2` demonstrate that GCondNet consistently enhances performance across different values of $K$. Notably, GCondNet greatly improves stability, yielding comparable outcomes across various network widths, unlike the MLP, which exhibits substantial performance decreases in narrower networks.
>
> `Changes:` We have added Appendix F.2, which includes this discussion, the complete numerical results varying $K$ and illustrative plots.
>
> *Table `R3` (same as Table F.8 in the updated manuscript): Test accuracy when varying the size of the first hidden layer of the backbone MLP model.*
>
> | Dataset | gli | gli | allaml | allaml | cll | cll | glioma | glioma |
> | --- | --- | --- | --- | --- | --- | --- | --- | --- |
> | Size first layer $K$ | MLP | GCondNet | MLP | GCondNet | MLP | GCondNet | MLP | GCondNet |
> | 20 | 75.38 ± 13.97 | 84.43 ± 10.84 | 78.18 ± 18.09 | 96.80 ± 5.57 | 67.97 ± 7.43 | 81.21 ± 7.53 | 66.00 ± 17.46 | 78.00 ± 14.26 |
> | 50 | 79.98 ± 10.44 | 84.02 ± 9.58 | 95.07 ± 7.39 | 97.58 ± 4.13 | 73.49 ± 4.66 | 81.29 ± 6.93 | 56.50 ± 16.93 | 74.50 ± 14.25 |
> | 100 | 77.72 ± 15.30 | 85.02 ± 9.00 | 91.30 ± 6.70 | 96.18 ± 4.90 | 78.30 ± 9.00 | 80.70 ± 5.50 | 73.00 ± 14.90 | 76.67 ± 12.90 |
> | 150 | 79.88 ± 15.32 | 84.92 ± 10.12 | 91.64 ± 10.22 | 95.16 ± 6.73 | 81.06 ± 7.82 | 80.60 ± 6.32 | 71.50 ± 15.67 | 67.33 ± 17.12 |
> | 200 | 81.03 ± 12.17 | 85.66 ± 10.04 | 91.89 ± 10.50 | 96.58 ± 4.73 | 80.11 ± 6.94 | 81.28 ± 6.37 | 74.33 ± 11.83 | 80.83 ± 11.60 |

---

> ### Author Response · Authors · 2024-06-21
> **Response by Authors**
>
> # Response-`nQny` (part 4/4)
> ## > W2: Clarity the definitions of “small” sample size and "large” feature number
>
> We follow the naming conventions from previous works (DNP [1], FSNet [2], WPFS [3]) that evaluated a similar scenario to ours, and although there is no standard definition, the term "small” datasets typically refer to having up to a few hundred samples, and "large” number of features to refer to thousands of features. Specifically, in our experiments, we assess 12 real-world tabular datasets with 72-200 samples and 3312-22283 features, many of which were used in related work [1, 2, 3], and which were referred as small-sample and high-dimensional datasets.
>
> We conducted two sets of new experiments demonstrating that GCondNet is applicable to datasets beyond small-size and high-dimensional data. First, we increased the dataset size, and the results indicate that GCondNet improves performance on larger datasets (as discussed in our response to RC2). Second, we varied the number of features, and the results show that GCondNet consistently enhances performance and stability across tasks with varying numbers of features (for more details, please see our response to Q1 for Reviewer x2re).
>
> `Changes:` We have updated the introduction and revised the example to clarify that "small" datasets refer to those with up to a few hundred samples, while high-dimensional data refers to those with thousands of features.
>
> Thank you again for your thoughtful feedback! We believe that the new results and discussion have improved the manuscript.

---

### Review · Reviewer_mcza · 2024-06-13

**Summary Of Contributions:**

In this article, the authors propose a new method to deal with high-dimensional datasets using graph neural networks (GNNs). When the number of dimensions, or features, exceeds the number of instances, the authors make use of this large dimensionality to train a GNN architecture that can generate feature-specific vectors on top of nearest neighbors graphs computed directly from the features. These vectors are then concatenated into a matrix, which itself serves as an initialization of some multilayer perceptron (MLP) architecture. At training time, these MLP parameters are computed as a linear combination of the GNN-based matrix and some standard, learnable parameter matrix. Moreover, the linear combination coefficient corresponding to the GNN-based matrix decreases to zero over iterations, so that the GNN-based matrix is not optimized and used after a while. The authors then demonstrate the usefulness of their approach by showing improvement (in terms of accuracy) over standard baselines on several high-dimensional datasets. They also show that such GNN-based vectors can also be incorporated to more general achitectures (such as transformers), that decaying the linear combination coefficient is indeed more efficient than keeping it fixed as it allows to avoid overfitting, and that GNN-based vectors are more efficient than other types of feature-specific vectors obtained with, e.g., NMF and PCA.

**Audience:**

Yes

**Claims And Evidence:**

Yes

**Requested Changes:**

1. Section 2.1: the construction of the kNN and SRD graphs is a bit unclear, as the distances used to build the graphs are not specified. I figured that these distances were simply obtained from the absolute differences between the feature values on the instances, but it is not clear from the text as it looks that the distances were computed out of the graph node features (which are one-hot encoding vectors), which does not make a lot of sense. This part should be more explicit.

2. The claim that decaying alpha reduces overfitting is well supported by Figure 3, but does not show so clearly from Table 1. I expected that the overfitting reduction would also induce smaller standard deviations on the accuracies but it seems that it is not the case (like on the glioma dataset). It would be good to add a discussion / explanation of this observation, as the results are currently a bit confusing w.r.t. that aspect.

3. About the results presented in Figure 3, the text mentions a test performance drop of 2% when fixing alpha, but I could not find the numbers in the figure (which only presents train and validation losses), nor in Table 1 (in which alpha is decayed). It would be nice to include the performance values in the text.

**Strengths And Weaknesses:**

Overall, I think this work is solid and interesting. The experiments are convincing and well-designed, with a good discussion over the different hyper parameters. Moreover, the paper is very clear and tackles a frequent and difficult problem in data science, namely the difficulty to handle high-dimensional datasets with neural networks. The weaknesses I identified were only about a few unclear points in the exposition, that I would like the authors to adress for the final version (see below).

---

> ### Author Response · Authors · 2024-06-21
> **Response by Authors**
>
> # Response-`mcza` (part 1/1)
>
> Thank you for your positive feedback and constructive suggestions! We have addressed all your comments below, and we have updated the manuscript with the new discussions and clarifications.
>
> ## > RC1: Clarify the distances for constructing the KNN and SRD graphs
>
> Indeed, we calculate distances using the absolute differences in feature values $X_{:, j}$ — and *not* on the one-hot-encoded graph nodes. For instance, if the feature values used to create one graph are $X_{:, j} = [\boldsymbol{x}^{(1)}_j, \boldsymbol{x}^{(2)}_j, \boldsymbol{x}^{(3)}_j]$, then the distances between nodes (based on which we create edges) are $\| \boldsymbol{x}^{(1)}_j - \boldsymbol{x}^{(2)}_j\|_1, \|\boldsymbol{x}^{(1)}_j - \boldsymbol{x}^{(3)}_j\|_1$, and $\|\boldsymbol{x}^{(2)}_j - \boldsymbol{x}^{(3)}_j\|_1$. While in this paper we use $\ell_1$ distance, other suitable distance functions can also be applied.
>
> `Changes:` We have updated Section 2.1 to clarify the distances used to construct the graphs.
>
> ## > RC2: Expected GCondNet to induce smaller standard deviations in Table 1 because it reduces overfitting
>
> GCondNet is a model which uses a backbone such as an MLP. Therefore, it is appropriate to compare the stability of GCondNet with training the standalone backbone model. Table 1 shows that GCondNet (with an MLP backbone) results in smaller average standard deviations than an MLP on the most extreme datasets—those with the smallest $N/D$ ratios. Specifically, on the five most extreme datasets, GCondNet reduces the average standard deviation by over 3.5% when using the SRD version and 2.5% when using the KNN version. On those extreme datasets, GCondNet also consistently outperforms an MLP with the same architecture and the most notable increases of 3-8%. As the $N/D$ ratio increases, GCondNet continues to surpass the baseline MLP, albeit with comparable standard deviations.
>
> To further explore GCondNet’s stability, we conducted additional experiments adjusting the $N/D$, keeping $N$ fixed and varying the number of features $D$. The new results in Table G.11 confirm the findings from Table 1, that GCondNet consistently enhances stability in the most extreme scenarios, reducing the standard deviation by up to 5%, and it converges to the backbone’s stability as the curse of dimensionality reduces. These findings underscore the role of GCondNet’s inductive bias and the alpha decaying mechanism in mitigating overfitting and enhancing stability, particularly in datasets with extreme $N/D$ ratios.
>
> Regarding the “glioma” dataset, Table 1 indicates that GCondNet achieves lower standard deviations—10.5% and 12.9% compared to 14.9% for the standalone MLP—alongside an accuracy improvement of up to 4.5%.
>
> `Changes:` We have updated Section 3.1 to include this discussion on GCondNet’s stability.
>
> ## > RC3: Add the numerical performance from Figure 3
>
> Figure 3 shows that fixing $\alpha$ results in a performance loss of at least 2% compared to the decaying $\alpha$ variant, which achieves 95.25% average test balanced accuracy. With fixed $\alpha \in \{0, 0.2, 0.4, 0.6, 0.8\}$, the performance ranges between 91.33% and 93.08%, with no particular trend. Training with a fixed $\alpha = 1$ results in a lower accuracy of 84.24%.
>
> `Changes:` We have updated Section 3.2 to include the numerical results from Figure 3.
>
> Thank you again for your thoughtful review! We believe that these clarifications have strengthened the manuscript.

---

### Decision · Action_Editor_a4q9 · 2024-07-29

**Recommendation:** Accept with minor revision

**Comment:**

Both the reviewers and the editors agree that the paper is interesting and relevant. The only request I have is that the authors carefully re-read their text and correct a few lingering typos and more carefully edit the text answered in response to the reviewer's comments.

**Audience:**

It is the editor's belief that the paper will be of interest to a sufficiently large TMLR audience.

**Claims And Evidence:**

The paper provides convincing and clear evidence in support of their results that improve the performance of NNs in the small-sample regime, with samples being high-dimensional. The key idea is to use side-information (implicit structures, patterns, correlations etc) present in such data. In my opinion, this application domain is of significance in biological data analysis. Also, all reviewers agree in their assessment of the claims made.